# Arctic cloud annual cycle biases in climate models

Patrick C. Taylor[1], Robyn C. Boeke[2], Ying Li[3] and David W.J. Thompson[3]

[1]NASA Langley Research Center, Climate Science Branch, Hampton, Virginia, USA

[2]Science Systems Applications Inc., Hampton, Virginia, USA

[3]Colorado State University, Department of Atmospheric Science, Fort Collins, Colorado, USA

*Correspondence to*: Patrick C. Taylor (Patrick.c.taylor@nasa.gov)

**Abstract.** Arctic clouds exhibit a robust annual cycle with maximum cloudiness in fall and minimum in winter. These variations affect energy flows in the Arctic with a large influence on the surface radiative fluxes. Contemporary climate models struggle to reproduce the observed Arctic cloud amount annual cycle and significantly disagree with each other. The goal of this analysis is to quantify the cloud influencing factors that contribute to winter-summer cloud amount differences, as these seasons are primarily responsible for the model discrepancies with observations. We find that differences in the total cloud amount annual cycle are primarily caused by differences in low, not high, clouds; the largest differences occur between the surface and 950 hPa. Grouping models based on their seasonal cycles of cloud amount and stratifying cloud amount by cloud influencing factors, we find that model groups disagree most under strong lower tropospheric stability, weak to moderate mid-tropospheric subsidence, and cold lower tropospheric air temperatures. Inter-group differences in low cloud amount are found to be a function of lower tropospheric thermodynamic characteristics. Further, we find that models with a larger low cloud amount in winter have a larger ice condensate fraction, whereas models with a larger low cloud amount in summer have a smaller ice condensate fraction. Stratifying model output by the specifics of the cloud microphysical scheme reveals that models treating cloud ice and liquid condensate as separate prognostic variables a simulate larger ice condensate fraction than those that treat total cloud condensate as a prognostic variable and use a temperature-dependent phase partitioning. Thus, the cloud microphysical parameterization is the primary cause of inter-model differences in the Arctic cloud annual cycle, providing further evidence of the important role that cloud ice microphysical processes play in the evolution and modeling of the Arctic climate system.

## 1. Introduction

Arctic clouds, arguably one of the most poorly understood aspects of the Arctic climate system, strongly modulate radiative energy fluxes at the surface, through the atmosphere, and to the top of the atmosphere (Cesana et al., 2012; Curry et al., 1996; Kay et al., 2008; Kay and L'Ecuyer, 2013; Shupe and Intrieri, 2004). As such, Arctic clouds have the potential to influence climate variability and change in the Arctic and globally. For instance, the presence of clouds

in winter over sea ice can be the difference between a -40 W m$^2$ surface radiative energy imbalance and a balanced
surface radiation budget, influencing surface temperature and sea ice growth (Morrison et al., 2012; Persson et al.,
2002, 2017). Accurately representing clouds in climate models is therefore necessary to realistically simulate the
evolution of the Arctic surface energy budget.
Contemporary climate models, however, strongly disagree with observations on the seasonality of Arctic cloud
radiative effects. Observations indicate that Arctic clouds cool the surface through the reflection of solar radiation for
a few months during summer and warm the surface through enhanced downwelling longwave radiation the rest of the
year (Kay and L'Ecuyer, 2013; Shupe and Intrieri, 2004). Climate models possess significant biases in the seasonality
of the surface cloud radiative effect (Boeke & Taylor, 2016; Karlsson & Svensson, 2013; Karlsson & Svensson, 2011).
Climate models participating in the Coupled Model Intercomparison Project 5 (CMIP5) (Taylor et al., 2011) simulate
Arctic clouds that are too reflective in summer and not insulating enough in winter. These cloud radiative effect biases
trace to a number of errors in cloud properties: namely, insufficient Arctic cloud amount (English et al., 2015),
inaccurate partitioning of cloud water between the liquid and ice phase leading to excessive ice clouds (Cesana et al.,
2012; Kay et al., 2016) and insufficient supercooled liquid clouds (Komurcu et al., 2014). This study focuses on errors
in model-simulated Arctic cloud amount and its annual cycle.
Arctic cloud amount exhibits a robust annual cycle that has been known for some time (Hahn et al., 1995;
Huschke, 1969). However, important revisions to our understanding of the cloud amount annual cycle have occurred
since the launch of the CloudSat Cloud Profiling Radar (Stephens et al., 2008) and the Cloud-Aerosol Lidar with
Orthogonal Polarization (CALIOP) (Winker et al., 2010). As illustrated in Liu et al., (2012), both ground observer
and satellite passive radiometer retrieval data sets indicate a broad summer maximum in cloud amount extending into
September, declining through fall, and reaching an annual cycle minimum in winter. Both data sets suffer from the
lack of sunlight in fall and winter. Passive cloud retrieval algorithms also change with surface type, posing additional
challenges (Minnis et al., 2011). CALIOP and CloudSAT active remote sensing instruments provide cloud amount
data independent of surface type with high accuracy in the absence of sunlight. Active remote sensing observations
indicate that average Arctic cloud amount exceeds 65% for each month reaching ~90% in fall (Boeke and Taylor,
2016; Liu et al., 2012) and that previous data sets missed ~10-15% of fall cloud cover. Space-based active retrievals
are not without limitations, most important of which is a 25-40% under-detection of clouds below 500 meters relative
to surface-based remote sensing observations (Liu et al., 2017). However, CALIOP and CloudSAT cloud amount data
still provide the most complete characterization of vertically-resolved Arctic-wide cloud amount.
Despite the refined observational knowledge of the Arctic cloud annual cycle, the mechanisms that control it
remain an open question. Beesley & Moritz (1999) outline several physical controls on Arctic clouds including
surface-atmosphere coupling, large-scale meteorology, and cloud microphysics. First, the surface-atmospheric
coupling mechanism implies—less sea ice and more surface evaporation—that Arctic cloud amount should follow the
annual cycle of sea ice. Observationally, this mechanism has been shown to operate under specific conditions in fall,
whereby reduced sea ice cover corresponds to increased cloud amount, but not in summer (Kay & Gettelman, 2009;
Morrison et al., 2018; Taylor et al., 2015). Second, seasonal changes in large-scale meteorology, atmospheric
advection, and humidity influence the cloud amount annual cycle. Previous work demonstrates a significant
dependence of cloud properties on local atmospheric conditions (Barton et al., 2012; Kay & Gettelman, 2009; Li et
al., 2014; Liu & Schweiger, 2017). Lower tropospheric stability has a profound influence on Arctic low cloud amount,
whereby increased stability corresponds to reduced cloud amount (Taylor et al., 2015). Third, cloud microphysical
processes affect cloud amount and exhibit a seasonality tied to temperature, whereby colder temperatures support ice
crystal formation and growth (e.g., via heterogeneous freezing and the Wegener-Bergeron-Findeisen process)
(Beesley and Moritz, 1999). The growth of ice crystals consumes available liquid, leading to precipitation. Once all
of the ice has fallen out, the atmosphere often transitions from a cloudy to clear state (Pithan et al., 2014). In addition,
the seasonality of aerosol amount and composition can influence cloud amount and properties by altering microphysics
(Coopman et al., 2018; Jackson et al., 2012).
Given the lack of mechanistic understanding of the drivers of the Arctic cloud annual cycle, it comes as no surprise
that climate models struggle to simulate the Arctic cloud amount annual cycle. Comparison of the CALIOP-CloudSAT
total column cloud amount with CMIP5 models indicates that individual models differ from observations by more
than 15% in summer and 40% in winter (Boeke and Taylor, 2016). Further, Boeke & Taylor, (2016) show that several
models produce peak cloud cover in winter whereas others producing peak cloud cover in summer; few models capture
the observed fall cloud cover peak. Thus, the majority of models misrepresent the annual cycle of Arctic cloud cover.
Meteorological reanalysis data products are not immune and also exhibit similar errors in the Arctic cloud amount
annual cycle timing (Liu & Key, 2016).
The combination of poor model simulation and the lack of mechanistic understanding of the drivers of the Arctic
cloud annual cycle signals a critical gap in our understanding with significant consequences for our ability to attribute,
simulate, and predict Arctic climate variability and change. We address this gap by investigating the drivers of the
inter-model differences in the Arctic cloud annual cycle in CMIP5 climate models. As indicated by previous studies,
Arctic cloud amount is influenced by its environment; a fact that guides this study. We adopt a methodology stratifying
climate model simulated vertically-resolved cloud amount by several key cloud influencing factors, described in
Section 2. The stratification methodology, discussed in Section 3, enables us to explore the dependence of simulated
cloud amount on individual and groups of cloud influencing factors and how they differ across the CMIP5 models. In
section 4, our key results are compared with previous work (Li et al., 2014a) and our understanding of the mechanisms
driving the Arctic cloud annual cycle is discussed. Lastly, Section 5 highlights the insights gained into how the Arctic
cloud annual cycle influences Arctic climate variability and change and our ability to simulate it.
**2. Methodology and Models**
The goal of this analysis is to explain the divergent representations of the Arctic cloud amount annual cycle found
in contemporary climate models. We use the historical forcing simulations (prescribed greenhouse gases and land use
changes consistent with observations from 1979-2005) from 24 CMIP5 climate models (Taylor et al., 2011; see Table
1 for a detailed description of each model and the corresponding model cloud and microphysics scheme). The model
outputs are available in the CMIP5 archive (https://esgf-node.llnl.gov/projects/cmip5/).
Several observed and reanalysis variables are included as a reference to gauge the fidelity of the model results.
The Modern-Era Retrospective Analysis for Research and Applications-2 (MERRA-2) provides information on the
Arctic atmospheric conditions. MERRA-2 has a horizontal resolution of 0.5° latitude x 0.625° longitude and vertical
resolution of 72 hybrid-eta levels fully described in Molod et al., (2015). The observed vertically-resolved Arctic
cloud amount are derived from CALIPSO-CloudSAT-CERES-MODIS (C3M) data (Kato et al., 2010). Vertical
profiles of cloud fraction are also included from ERA-Interim reanalysis (Dee et al. 2011).
The primary methodology composites cloud amount into bins of individual cloud influencing factors, adapted
from Li et al., (2014). The cloud influencing factors considered include vertically-resolved cloud amount, air
temperature ($T_s$), relative humidity (*RH*), 500 hPa vertical velocity ($\omega_{500}$), sensible heat flux (*SHF*), latent heat flux
(*LHF*), liquid and ice water mixing ratios (*CLW* and *CLI*, respectively), sea ice concentration (*SIC*) and lower
tropospheric stability (*LTS*). Lower tropospheric stability is defined as the potential temperature difference between
the surface and 700 hPa, computed from the monthly-averaged temperature profile. We also extend our composite
analysis beyond single variables and construct joint distributions.
The primary difference between the present analysis and Li et al., (2014) is the use of monthly-averaged model
output instead of instantaneous satellite data. To understand the potential shortcomings of using monthly-averaged
output instead of daily output calculations were investigated by carrying out the analysis using daily data based on
one available model (IPSL-CM5A-LR). The results (not shown) indicated that the largest difference between using
daily and monthly mean model output was due to the lesser dynamic range on monthly timescales. Overall, the daily
and monthly mean results agree in the most frequently occurring meteorological conditions. The largest differences
between the daily and monthly results occur in winter for high stability regimes (*LTS* > 34) in which daily data shows
about 10% larger *CA* than monthly; however, these regimes occur with a frequency less than 0.1%. We also note that
the covariances between clouds and cloud influencing factors evaluated at daily and monthly timescales represent
different manifestations of processes; thus, different processes may be important for explaining cloud behavior and
model differences at the daily and monthly timescales. As such, care must be taken in the interpretation of the results
at monthly timescales. We do not expect that the use of monthly averaged data to affect the main conclusions, however
an analysis performed at the daily timescale provides more detailed information due to the larger dynamic range with
the potential to identify additional processes that cause model differences under the wider range of atmospheric
conditions.
Lastly, the results are composited and analyzed within two groups based upon key features of the simulated Arctic
total cloud amount annual cycle. Figure 1a shows that the cloud amount annual cycles from individual models tend to
follow one of two patterns: 1) largest cloud amount in winter with small seasonal variations, and 2) minimum cloud
amount in winter and maximum cloud amount in summertime/early autumn, with large seasonal amplitude. Figure 2
further summarizes these two patterns by showing a scatterplot of the average winter (DJF) and summer (JJA) cloud
amounts for individual models. This result motivates the separation of the 24 models into two groups: models that
simulate a larger total cloud amount in winter are referred to as Group 1 (10 models), whereas models that simulate a
larger total cloud amount in summer are referred to as Group 2 (14 models).
While the models can be grouped in several different ways, the choice to delineate model groups above and below
the diagonal 1:1 line in Fig. 2 clearly places models with similar cloud amount annual cycle shapes together while
also grouping them based on how they differ from C3M observations and two reanalyses (see stars in Fig. 2). Group
1 models show maximum cloud amount in winter, which closely resemble MERRA-2 but differ from C3M
observations. Group 2 models correctly simulate the winter-season minimum cloud amount, consistent with C3M, but
possess 1) a much larger-amplitude of annual cycle than that in either C3M or reanalysis and 2) a summer peak in
cloud amount as opposed to fall, as seen in both C3M and ERA-Interim. This separation is also motivated by the need
to understand the factors (e.g., microphysics, surface turbulent fluxes, dynamics, and thermodynamics) responsible
for producing clouds in these individual seasons and to provide insight as to the cause(s) of the differences in Arctic
cloud amount annual cycle between models. The application of this grouping allows us to consolidate the analysis and
take a deeper look at the influencing factors.
As a test of the robustness of the grouping strategy, we created a third group containing the five models closest
to the C3M observations (hereafter Group 3; bcc-csm1-1, CMCC-CM, CanESM2, MPI-ESM-MR, and MPI-ESM-
LR). Composites of $CA$ for from Group 3 show features present in both Group 1 and Group 2, as expected since Group
3 contains models from each (not shown). This indicates that even the models closest to observations display features
from their respective group. If the 1:1 line was a poor metric to use for group selection, we would expect Group 3 to
resemble one of the groups or neither of the groups. Thus, the results are robust to a small change in the grouping
strategy.
**3.   Results**
**3.1.  Vertical variations of the cloud amount annual cycle**
Figure 3 illustrates the vertically-resolved average cloud amount annual cycle for each model group observations
(bottom panels). Observations and two reanalyses (Fig. 3g-h) all agree on the timing of minimum low cloud amount
during summer. The peak season of the low cloud amount is slightly different. For example, both C3M and ERA-
Interim (Figs. 3g,h) show the peak in low cloud amount and vertical extent in late autumn around October, whereas
the MERRA-2 reanalysis (Fig. 3f) shows the low cloud amount peak in winter around January and February.
Group 1 (Fig. 3a) exhibits a minimum in low cloud amount (>850 hPa) in May through July with a maximum
low cloud amount in January and February. Group 1 high cloud amount follows a similar seasonal pattern as low
clouds with a minimum in summer and maximum in the fall/winter at reduced amplitude. Group 2 (Fig. 3b) exhibits
a similar high cloud amount annual cycle as Group 1 with smaller cloud amounts and a weaker amplitude. However,
the annual cycle of low cloud in Group 2 indicates that cloud amount slowly increases in amount and extends in height
through summer, then sharply decreases after September, in sharp contrast with C3M observations, MERRA-2
reanalysis Group 1 (Fig. 3f,g) and Group 1 (Fig. 3a).
The standard deviation in cloud amount across each group (Fig. 3d,e) indicates that the intra-group differences
are greatest in the lowest levels of the atmosphere during all months for both groups. Specifically, the standard
deviation in cloud amount is greatest at vertical levels and times of year with the largest cloud amount, below 800 hPa
and above 500 hPa in winter for both groups and below 800 hPa in summer. The only exception is in Group 1 where
larger standard deviations occur in summer below 800 hPa, when Group 1 models show minimum cloud amount.
The seasonal cycle of the vertically-resolved cloud amount (Fig. 3) are consistent with the results in Figs. 1b,c,
which illustrate the simulated and observed seasonal cycles of Arctic cloud amount for low clouds (1000-850 hPa)
and high clouds (500-300 hPa), respectively. The results in Figs. 1a,c demonstrate that low clouds predominantly
contribute to the winter versus summer peaks in the simulated seasonal cycle of the total cloud amount. The rest of
this paper analyzes how the dependence of cloud amount on the cloud influencing factors contributes to these
differences in Arctic low cloud amount in winter versus summer. The goal of this paper is to understand how, why
and to what extent do the cloud influencing factors contribute to the differences in the Arctic low cloud amount with
winter peaks in Group 1 and late summer peaks in Group 2.

## 3.2. Horizontal variation in the cloud amount annual cycle

The above differences in the annual cycle of the Arctic clouds between Groups 1 and 2 are based on the averages
over the entire Arctic region, in this subsection we further confirm that such differences in sign are spatially uniform
across the Arctic. Figure 4 illustrates the spatial variations of the low and high cloud amount differences for Group 1
minus Group 2. In winter, Group 1 produces an average of 12.4% more low clouds than Group 2 (Fig. 4a) and 7.3%
fewer low clouds in summer (Fig. 4c). These differences are generally spatially uniform. Differences in high cloud
amount show similar spatial uniformity but with Group 1 producing more high clouds than Group 2 in both winter
(+6.4%) and summer (+3.7%) (Fig. 4b,c). Overall, the spatial variability of the difference is very weak (i.e, the
differences in the average high and low cloud amount between land, ocean, and all surface types are generally less
than 1%); thus, regional differences do not significantly contribute to the annual cycle differences in low or high cloud
amount.
Since atmospheric and surface properties vary across the Arctic and can influence the simulated cloud amount,
we also analyze the spatial variations in cloud influencing factors for the model groups (not shown) and find that the
differences between Group 1 and 2 exhibit a general spatial uniformity with only minor deviations. As such, the
following stratification analysis is performed over the entire Arctic region.

## 3.3. Inter-group differences in mean and distribution of atmospheric conditions

Arctic cloud formation is influenced by a number of atmospheric characteristics including surface and boundary
layer thermodynamic properties and large-scale dynamics (Kay & Gettelman, 2009; Z. Liu & Schweiger, 2017; Taylor
et al., 2015). Table 2 and Figure 5 provide the annual-mean ensemble averages of cloud influencing factors for each
group and their probability density functions (PDFs) over the ocean and land surfaces. Although the average properties
for all cloud influencing factors between the two groups are significantly different at 95% confidence (fourth column

in Table 2), the differences are generally very small, suggesting that differences in the average atmospheric conditions do not drive inter-group differences in the cloud amount annual cycle. Notable differences found for *LTS*, *RH* and *CLW* over both surface types, with the values in Group 2 higher than those in Group 1. Overall, the spread in the average cloud influencing factors is larger within each group than between Group 1 and 2.

The variability of individual cloud influencing factors is consistent between the groups with some small differences. The PDFs in Fig. 5 summarize the frequency of the cloud influencing factors for Group 1 (red) and Group 2 (blue) separated into land (cross-hatching) and ocean (solid). Figure 5 includes PDFs of each variable derived from MERRA-2 reanalysis with solid black lines for ocean (square symbols) and land (triangle symbols). In most cases, the distribution of cloud influencing factors is similar between the two groups for each surface type. Consistent with Table 2, the most notable differences between the groups are (1) Group 2 models exhibit a higher frequency of stronger *LTS* values for both land and ocean (Fig. 5a) and (2) Group 2 $-\omega_{500}$ exhibits a higher frequency of values near 0 hPa day$^{-1}$ over both land and ocean (Fig. 5b). In these cases, Group 1 $-\omega_{500}$ and *LTS* is more consistent with MERRA-2. Additional group differences are seen in *RH* (Fig. 5g), *CLI* (Fig. 5d) and *CLW* (Fig. 5h) whereby Group 2 favors higher *RH*, larger *CLW*, and a higher frequency of *CLI* values near 0 g kg$^{-1}$ while Group 1 shows a higher frequency of *CLW* values near 0 g kg$^{-1}$.

**3.4. Dependence of vertically-resolved cloud amount on cloud influencing factors**

We investigate the extent to which inter-group differences in cloud amount are explained by differences in the relationship between cloud amount and cloud influencing factors. Figure 6 shows the vertically-resolved average cloud amount in DJF binned by five different cloud influencing factors: $-\omega_{500}$, *LTS*, ice water path (*IWP*), total condensed water path (*CLWVI*; ice plus liquid), and *SIC*, all of which show relatively large inter-group differences as compared to other variables (see Table 2 and Fig. 5). Since Group 1 models show a winter cloud amount peak in the annual cycle, it is expected that Group 1 produces larger cloud amounts than Group 2 throughout the troposphere and especially below 850 hPa for most cloud influencing factors (Fig. 6, right column).

Figure 6a,b illustrates the cloud vertical structure in DJF as a function of $-\omega_{500}$ and reveals a general increase in cloud amount as the strength of rising motion increases at most levels for both groups over ocean (from left to right in Fig. 6a,b) and land (Fig. S1). Note that for levels >950 hPa, cloud amount in Group 1 exhibits larger cloud amounts under both sinking and rising motion, and also contributes to large inter-group differences at pressures >950 hPa (Fig. 6c).

Figure 6d,e illustrates the dependence of the vertically-resolved cloud amount in DJF stratified by *LTS*. Both groups exhibit a general decrease in cloud amount and its vertical extent with stronger *LTS* at all levels and over both ocean and land (Fig. S1); in other words, as conditions become more stable clouds tend to occur less frequently and are constrained to a shallower layer closer to the surface, also found in observations (Taylor et al., 2015). Much like $-\omega_{500}$, Group 1 produces equal or larger cloud amounts at pressures >950 hPa as *LTS* increases, signaling a potentially important $-\omega_{500}$-*LTS* covariance (discussed below). Specifically, the average cloud amount is >20% larger

in Group 1 than in Group 2 when *LTS* > 20 K at pressures >950 hPa. The larger cloud amounts at pressures >950 hPa in Group 1 can be viewed as either a difference in a dissipative mechanism (e.g., turbulent mixing, cloud microphysics, or precipitation) between the groups or a difference in cloud production (e.g., ice formation or surface buoyancy).

Figure 6g,h,j,k illustrates the dependence of cloud amount in DJF on *IWP* and *CLWVI*. Models in both groups favor more cloud amount with higher cloud bases for increasing *IWP* and *CLWVI*; both surface types exhibit similar behavior. Group 1 diverges from Group 2 at lower values of *IWP* and *CLWVI* (< ~35 g m$^2$) by producing maximum cloud amount in the thin cloud regime at pressures >950 hPa (Fig. 6g,j) while Group 2 shows minimum cloud amount. For the average wintertime values of *IWP* (~32 g m$^2$) and *CLWVI* (~52 g m$^2$), Group 1 has larger cloud amount than Group 2 at all levels over ocean and land.

 The influence of surface conditions on cloud amount over the Arctic Ocean is assessed using *SIC* (Fig. 6m-o). Representing an integral measure of the surface influence on cloud amount, increased *SIC* generally corresponds to decreases in surface turbulent fluxes and stronger *LTS* (Pavelsky et al., 2011; Taylor et al., 2018). Figures 6m,n illustrate that both groups produce a decrease in cloud amount and lower cloud bases with increased *SIC* although the cloud amount is higher in Group 1 than in Group 2. As for other variables, this relationship is weakened in Group 1 at pressures >950 hPa.

Figure 7 shows the vertically-resolved average cloud amount dependence on four different cloud influencing factors (-$\omega_{500}$, *LTS*, *IWP*, and *CLWVI*) over land and one (*SIC*) over ocean for summer (JJA). We show results over land in summer because differences exceed 20% over land and are 5-10% over ocean. Since Group 2 includes models with a summer cloud amount peak in the seasonal cycle (especially for low clouds), it is expected that Group 2 models generally produce larger cloud amount than Group 1 throughout the troposphere for almost all cloud influencing factors (right column). The largest inter-group differences are again at pressures >950 hPa.

Important findings from Fig. 7 include (1) the inter-group differences in cloud amount are ~5-10% smaller during summer, (2) Group 2 tends to produce more clouds at pressures >950 hPa for all cloud influencing factors, (3) all dependencies of cloud amount on cloud influencing factors are weaker in summer than in winter, and (4) neither group exhibits a dependence of the average cloud fraction on *SIC*. Only cloud amount dependencies with -$\omega_{500}$, *IWP,* and *CLWVI* illustrate a noteworthy gradient in summer where Group 2 produces a stronger low cloud amount increase as rising motion increases and at larger *IWP*/*CLWVI* values.

The winter and summer analyses reveal several key takeaways. First, the primary inter-group differences are found at pressures >950 hPa in the thin, low ice cloud regime (IWP< 35 g m$^2$) in winter and the thicker low cloud regime (IWP > 70 g m$^2$) in summer. Second, the differences in the cloud amount dependence on cloud influencing factors are larger during winter than summer. Third, the largest inter-group differences are found under stable conditions (LTS > 20 K) and sinking motion in winter and under rising motion in summer. The fact that inter-group differences in the cloud amount dependence are largest for *LTS* and -$\omega_{500}$ and the expectation of significant covariances

between these two variables warrants simultaneous analysis using a joint distribution to address the question, why are
Group 1 models able to maintain large low cloud fraction under strong stability and subsidence?

**3.5. Joint PDFs: *LTS* and *-ω₅₀₀***

Figure 8 shows the joint distribution of average low cloud amount in winter stratified by both *LTS* and $-\omega_{500}$ (Fig.
8a-b), and superimposed with the corresponding frequency of occurrence (contours) for Group 1 (Fig. 8a) and Group
2 (Fig. 8b). Cloud amount depends on both (1) the relationship between the cloud amount and *LTS* and $-\omega_{500}$ and (2)
how frequently each LTS and $-\omega_{500}$ bin occurs. For regions with *LTS* < 12 K, low cloud amount for both groups is
primarily a function of *LTS* with little dependence on $-\omega_{500}$; the inter-group differences illustrate the same behavior
(Fig. 8c). Considering *LTS* >12 K, low cloud amount exhibits a dependence on both *LTS* and $-\omega_{500}$, however the inter-
group differences (Fig. 8c) still correspond only to variations in *LTS*.
While both groups simulated the highest frequency of occurrence of $-\omega_{500}$ bin around -4 hPa day⁻¹, Group 1 most
frequently simulates LTS values between 22-24 K whereas Group 2 simulates slightly higher values between 26-30
K (Fig. 8a,b contours). Thus, the inter-group difference is marked by a dipole pattern along the LTS axis between 22-
24 K and 26-30 K, and these regions contribute most to the winter low cloud amount between Group 1 and Group 2.
Figure 9 shows the joint distribution of low cloud amount by *LTS* and $-\omega_{500}$ bins and the corresponding frequency
of occurrence in summer. The pattern in the summer average low cloud amount is more similar between the groups
(Figs. 9a,b) compared to winter yielding smaller inter-group differences (Fig. 9c). For *LTS*<14 K, low cloud amount
depends primarily on *LTS* with a weak dependence on $-\omega_{500}$; whereas for *LTS*>14 K, low cloud amount depends on
both *LTS* and $-\omega_{500}$, a behavior similar to winter. Additionally, the low cloud amount gradients are sharper in summer
than winter, meaning that summer low cloud amount is more susceptible to small changes in *LTS* and $-\omega_{500}$ than in
winter. The inter-group differences in frequency of occurrence indicates that Group 2 exhibits higher *LTS* values (20-
25 K) and lower *LTS* values (<12 K) more frequently than Group 1.
Based on the relationships between low cloud amount and $LTS_i$ and $-\omega_{500,j}$ as illustrated in Figs. 8 and 9, the winter
or summer average low cloud amount can be estimated using

$$\overline{LCA} = \sum_{i,j} LCA\left(LTS_i, -\omega_{500,j}\right) * RFO\left(LTS_i, -\omega_{500,j}\right). \tag{1}$$

This expression describes the weighted sum of the low cloud amount over all $LTS_i$ and $-\omega_{500,j}$ from each $i, j$ bin, where
$LCA\left(LTS_i, -\omega_{500,j}\right)$ corresponds to the low cloud amount as a function of $LTS_i$ and $-\omega_{500,j}$ and $RFO\left(LTS_i, -\omega_{500,j}\right)$
corresponds to the relative frequency of occurrence of each $LTS_i$ and $-\omega_{500,j}$ bin. Applying Eq. (1) to compute the average
low cloud amount, $\overline{LCA}$, in either winter or summer reproduces the winter and summer average low cloud amount for
each group to within 1-2% percent (Table 3). We construct $LCA\left(LTS_i, -\omega_{500,j}\right)$ by averaging across months and
models, thus removing some variability. As such, eq. (1) parameterizes low cloud amount and is not expected to
exactly reproduce $\overline{LCA}$ . This exercise indicates that $\overline{LCA}$ can be accurately reconstructed using the
$LCA(LTS_i, -\omega_{500,j})$ and $RFO(LTS_i, -\omega_{500,j})$ suggesting that this approach is applicable to interpreting drivers of
interannual variability or feedbacks in low cloud amount.
Equation (1) can be applied to both Group 1 and Group 2, and then the inter-group differences (Group 1 minus
Group 2; $\overline{\delta LCA_{G1-G2}}$) can be estimated and decomposed using a first-order Taylor series approximation to further
quantify the relative contributions from differences in 1) $\delta LCA(LTS_i, -\omega_{500,j})$ and 2) $\delta RFO(LTS_i, -\omega_{500,j})$.
$$\overline{\delta LCA_{G1-G2}} = \sum_{i,j} \left[ \left( \delta LCA(LTS_i, -\omega_{500,j}) \right)_{G1-G2} * RFO(LTS_i, -\omega_{500,j})_{G1} \right) \right] +$$

$$\sum_{i,j} \left[ \left( LCA(LTS_i, -\omega_{500,j})_{G1} * \delta RFO(LTS_i, -\omega_{500,j})_{G1-G2} \right) \right] \qquad (2)$$

In Eq. (2), $\overline{\delta LCA_{G1-G2}}$ corresponds to the inter-group difference (Group 1 minus Group 2) in average low cloud
amount, $\delta LCA(LTS_i, -\omega_{500,j})_{G1-G2}$ corresponds to the inter-group difference in the dependence of low cloud amount
on $LTS$ and $-\omega_{500}$, and $\delta RFO(LTS_i, -\omega_{500,j})_{G1-G2}$ corresponds to the inter-group difference in the relative frequency
of occurrence of $LTS$ and $-\omega_{500}$ bins. In this framework, the first term on the right-hand side represents the influence of
the parameterized cloud physics (due to $\delta LCA(LTS_i, -\omega_{500,j})_{G1-G2}$) and the second term represents the influence of
atmospheric state occurrence (due to $\delta RFO(LTS_i, -\omega_{500,j})_{G1-G2}$). Table 4 summarizes the results indicating that the
$\delta LCA(LTS_i, -\omega_{500,j})_{G1-G2}$ term is responsible for the summer and winter inter-group differences in low cloud amount.
While this result attributes the Group 1 minus Group 2 differences to parameterized cloud physics and not the
atmospheric state occurrence, it does not explain the fundamental cause. The cause(s) is due to differences in the
specifics of the parameterized cloud physics, systematic differences in the atmospheric conditions within $LTS$ and $-\omega_{500}$
bins, or a combination of both. A systematic exploration of the inter-group differences in cloud physics
parameterizations are beyond the scope of this study. However, we explore the inter-group differences in atmospheric
conditions within $LTS$ and $-\omega_{500}$ bins and perform an additional stratification based upon specifics of the cloud
microphysical schemes (Table 1) to assess the influence on low cloud amount differences.
Characterizing atmospheric state by $LTS$ and $-\omega_{500}$ bins does not account for all inter-group differences in
atmospheric state. Thus, we consider atmospheric and surface conditions stratified by $LTS$ and $-\omega_{500}$ (Fig. 10). Both
groups exhibit similar distributions of lower tropospheric $RH$, 950-hPa $T_a$, $SHF$, $LHF$, and $SIC$ (not shown) within the
$LTS$ and $-\omega_{500}$ bins in winter (Fig. 10) and summer (Fig. S3). Inter-group differences in $RH$ (Fig. 10c) are generally
<5% and anti-correlate with inter-group low cloud amount differences; in other words, Group 2 exhibits smaller low
cloud amount than Group 1 and yet has a larger $RH$ and more frequently simulates values >80% (Fig. 5g).
Alternatively, Group 1 is colder than Group 2 in the most frequently occurring bins (Fig. 10f), suggesting differences
in cloud microphysics and ice formation. Inter-model differences in $SHF$ and $LHF$ indicate that the inter-group
differences change sign with increasing $LTS$; however, these differences anti-correlate with the differences in low
cloud amount.

Inter-group differences in cloud microphysics and specifically the production of cloud liquid versus ice strongly corresponds to inter-group differences in low cloud amount. Figure 11 illustrates the differences in winter lower tropospheric *CLW* (Fig. 11 a-c), *CLI* (Fig. 11 d-f), and ice condensate fraction, (*ICF*; Fig. 11 g-i) stratified by *LTS* and $-\omega_{\omega}$. ICF is defined as the ratio of *CLI* and *CLWVI*. Results for summer are presented in Figure 12. Both groups exhibit similar overall dependencies of the liquid and ice water mixing ratio on *LTS* and $-\omega_{\omega}$ with Group 2 producing more cloud liquid than Group 1 (Fig. 11c) and slightly more cloud ice (Fig. 11f). The ICF (Fig. 11g,h), however, indicates that Group 1 produces a much higher percentage of total condensate as ice (*ICF* greater than 0.5 in the most frequently occurring regimes). Figures 11 and 12 support the idea that Group 1 models sustain a larger fraction of thin ice clouds at cold temperatures supporting larger low cloud amount in winter. Moreover, the finding that Group 1 models are drier than Group 2 suggests that the enhanced cloud ice formation dehydrates the winter Arctic atmosphere. The smaller *CLW* in Group 1 may also be related to the greater *CLI* as some models do not allow supersaturation with respect to ice meaning that liquid supersaturation would not be reached under most Arctic winter conditions. This result is consistent with Kretzschmar et al., (2018) showing that not allowing ice supersaturation corresponds to a positive bias in low cloud cover in ECHAM6. Alternatively, the larger cloud liquid production by Group 2 corresponds to a larger low cloud amount in summer. The results support the argument that cloud phase partitioning and cloud microphysical parameterizations explain the differences in the Arctic cloud amount annual cycle and differences in the surface turbulent fluxes and atmospheric circulation contribute little. Therefore, improved representation of the Arctic cloud amount annual cycle requires improvements in the representation of cloud microphysical processes especially in thin, low clouds.

To further investigate the role of microphysics, we first set out to stratify the models into new groups based upon whether or not supersaturation with respect to ice was allowed. However, we were not able to and found that the required information about to whether a particular model allows ice supersaturation or not is not consistently identified in the citing literature (Table 1). Sufficient detail is provided in the literature to partition the models into Group A those that treat cloud ice and water as prognostic variables and Group B those that treat total water as a prognostic variable and use a temperature-dependent phase partitioning. Figure 13 illustrates the joint distributions of low cloud amount, CLW, CLI, and ICF in DJF. While Groups A and B both contain Groups 1 and 2 models, the distributions of CLW, CLI, and ICF in Fig. 13 resembles that shown in Fig. 11. The results indicate that models treating total cloud water as a prognostic variable and use a temperature-dependent phase partitioning have a smaller ICF (less cloud ice and more cloud water) than those that treat cloud ice and liquid as separate prognostic variables. The cloud fraction differences between this microphysical scheme-based grouping is smaller than the original group but also takes on the same shape. Thus, the cloud microphysical treatment is a principle factor explaining the differences in the inter-group low cloud amount differences.

Due to the importance of $T_{A}$ and *RH* to this explanation, we further investigate the low cloud amount dependence on $T_{A}$ and *RH* as both variables influence the cloud microphysics parameterizations. Figures 14 and 15 illustrate the joint distribution of the average low cloud amount stratified by lower tropospheric $T_{A}$ and *RH* and frequency of occurrence of each bin in winter and summer, respectively. The largest inter-group differences are found at the coldest

temperatures and highest *RH* values for both winter (Fig. 14) and summer (Fig. 15). Group 1 favors cooler and drier
atmospheric conditions than Group 2 (Fig. 14c), while also producing more clouds under those conditions. In summer,
Group 2 models produce larger low cloud amounts compared to Group 1 in the warmer and more humid conditions
occurring most frequently (Fig. 15). Group 2 also slightly favors more humid conditions in summer than Group 1
contributing to larger summer low cloud amount. Results applying the decomposition from (1) to the $T_A$ and *RH* joint
distribution indicate that in winter differences in the parameterized cloud physics are primarily responsible for
$\delta LCA_{G1-G2}$, where as in summer the relative frequency of occurrence is primarily responsible for $\delta LCA_{G1-G2}$ (Table
4). This result supports our conclusion that cloud microphysical processes explain the model differences in Arctic low
cloud amount in winter. In summer, however, Fig. 15 indicates that processes that control the frequency of occurrence
of $T_A$ and *RH* states are also important to explain low cloud amount differences.
**4.   Discussion**
This analysis explores the factors that influence Arctic cloud amount within contemporary climate models with
the specific focus on understanding the factors that drive differences in the simulated Arctic cloud amount annual
cycle. In comparing our results with previous work, the vertically-resolved cloud amount dependencies (Figs. 6 and
7) on cloud influencing factors agree with the observationally-based analysis of Li et al., (2014). It should be noted
that this result is despite differences in the temporal characteristics of the two analyses: monthly-averaged model
output vs. instantaneous satellite data. This result suggests that the use of monthly averages is not as big of a limiting
factor for investigating the cloud dependence on atmospheric and surface conditions as previously assumed. Our
results demonstrate that climate model parameterizations realistically reproduce the general Arctic cloud amount
dependence on atmospheric conditions, yet subtle differences produce large discrepancies in the Arctic cloud amount
annual cycle between models and between models and observations. While a thorough model-observation comparison
using CALIPSO-CloudSAT satellite simulator output is the subject of ongoing work, our results indicate that neither
Group 1 or 2 reproduces observations (Fig. 3). Individual models significantly outperform the Group 1 and 2 averages
as indicated by the close proximity of five models (bcc-csm1-1, CMCC-CM, CanESM2, MPI-ESM-MR, and MPI-
ESM-LR) to the observations (denoted by stars) in Fig. 2.
We argue that the primary cause of the larger cloud amount in Group 1 during winter is due to the production and
maintenance of low, thin ice clouds at colder surface air temperatures than Group 2. We hypothesize that Group 1
maintains low cloud amount at colder temperatures as a result of cloud microphysical parameterization differences
that produce a larger fraction of cloud ice than Group 2 overall and especially at colder temperatures and lower *RH*
(Fig. S4 and S5 illustrates the *ICF* stratified by *RH* and $T_A$). This hypothesis seems at odds with previous cloud process
studies considering the mixed-phase cloud system where high cloud ice production desiccates super cooled liquid and
more efficiently precipitates reducing low cloud amount (Avramov et al., 2011; Morrison et al., 2012). In this case,
the results suggest that Group 1 overcomes this by producing more cloud ice and not by not precipitating the ice out
of the atmosphere. In addition, we do not know the frequency of mixed-phase clouds from monthly averaged output.
Overall, the importance of cloud microphysics to model cloud amount is consistent with previous work illustrating
that Arctic clouds and their radiative effects strongly respond to changes in ice microphysics (English et al., 2014;
Kay et al., 2016; McCoy et al., 2016; Pithan et al., 2014; Tan and Storelvmo, 2015).

What do our results argue about the drivers of the Arctic cloud annual cycle? The climate model results argue
that the Arctic cloud annual cycle is most strongly driven by the seasonality of cloud microphysics, specifically the
cloud phase and temperature relationship. The *SIC* in both the inter-group differences as well as the cloud amount
dependence on sea ice shows a weaker relationship than other factors indicating a limited role in driving the Arctic
cloud annual cycle. The results do not support a significant role for the seasonality of relative humidity in forcing the
Arctic low cloud annual cycle because (1) the seasonality of *RH* is similar between the two groups (Fig. S3) and (2)
models that produce fewer winter clouds possess higher *RH*. Rather, the cloud microphysics appear to shape Arctic
lower tropospheric *RH*. Changes in atmospheric conditions, specifically *LTS* and $-\omega_{500}$, are significant between winter
and summer indicating a role for the large-scale circulation. Our results support the idea of Beesley & Moritz (1999)
that the covariance between atmospheric temperature and cloud microphysics is a major factor responsible for the
Arctic cloud annual cycle.

A critical consideration is the cloud ice formation process. Models that do not allow supersaturation with respect
to ice implicitly assume that deposition freezing is the dominant ice formation process in Arctic low clouds. However,
observational evidence indicates that supercooled liquid must first be present before cloud ice is observed at
temperatures warmer than -25◦C, supporting the notion that immersion freezing is the dominant ice nucleation process
(de Boer et al., 2011). Our results indicate that a better understanding of ice formation mechanisms operating in the
Arctic and the conditions under which each dominates would provide an important constraint on climate model physics
and Arctic climate simulations. Moreover, additional model studies like (Kretzschmar et al., 2018) that investigate the
influence of ice supersaturation on Arctic low cloud amount are needed.

A new idea from this analysis is one of Arctic cloud susceptibility. Returning to the *LTS* and $-\omega_{500}$ joint
distributions, summer versus winter differences (Figs. 8a,b, and 9a,b) in the low cloud amount dependence are
significant. Figures 8 and 9 show that the most frequently occurring atmospheric conditions in summer are found
along a strong gradient in the low cloud amount dependence on *LTS* and $-\omega_{500}$, not the case for winter. This suggests
that summer low cloud amount is more susceptible to changes in atmospheric conditions than winter low clouds. This
apparent difference in the susceptibility of low cloud amount to changes in atmospheric conditions could have
important implications for Arctic cloud feedback, as (Taylor, 2016) illustrates that changes in *LTS* imply large changes
in the surface cloud radiative effect.
**5.  Conclusion**

Surface and space-based observations of Arctic clouds exhibit a robust annual cycle with maximum cloud amount
in fall and a minimum in winter. Variations in cloud amount affect energy flows in the Arctic and strongly influence
the surface energy budget. Therefore, understanding the role of clouds in the context of the present-day Arctic climate
is imperative for improving predictions of surface temperature and sea ice variability, as well as for projecting Arctic
climate change. As we and several authors before demonstrate, contemporary climate models struggle to reproduce
observed Arctic cloud amount and its variability, especially within the context of the annual cycle.
Our analysis focuses on identifying the causes of the climate model differences in the annual cycle representation.
We find that most climate models tend to fall into one of two groups: one favoring larger winter cloud amount and
another favoring larger summer cloud amount. The results demonstrate that differences in low, thin ice clouds at
pressures >950 hPa, not middle or high clouds, are primarily responsible for the total cloud amount annual cycles
within each group. These discrepancies between the two model groups exhibit little spatial variability, are consistent
between land and ocean, and are only weakly influenced by sea ice concentration, suggesting that the cause of the
cloud amount differences operates Arctic-wide.
Differences in atmospheric and surface conditions represent an important potential source of the low cloud
amount differences. The results show small differences in the annual, domain-averaged atmospheric and surface
conditions between the two groups and indicate that these are not responsible for the low cloud amount differences.
Considering specific atmospheric and surface conditions, we find that models disagree most under strong lower
tropospheric stability, weak to moderate mid-tropospheric subsidence, and cold lower tropospheric air temperatures.
Overall, the cloud amount dependence on cloud influencing factors explains most of the inter-group differences in
cloud amount. Since, the cloud amount dependence on cloud influencing factors in climate models is governed by
parameterized cloud physics, the results indicate that parameterization differences are responsible for the cloud
amount discrepancies and that differences in the frequency of occurrence of atmospheric and surface conditions
between the models is not a significant factor.
Why do models simulate different low cloud amounts under specific atmospheric conditions? Models produce
similar dependencies of low cloud amount on atmospheric and surface conditions in summer but not in winter. Models
able to sustain larger low cloud amounts at colder surface air temperatures simulate more winter clouds and we argue
that the details of the cloud microphysical parameterization are responsible by maintaining a larger fraction of cloud
ice in some models than others. The present analysis is unable to isolate the specific characteristics of the ice
microphysical parameterization (e.g., ice formation, crystal habit, mass-diameter relationship, fall speed, gamma size
distribution parameters, etc.) that drive these differences, however this should be the focus of future investigation. A
commonality of these ice microphysical parameterization characteristics is that few observational constraints are
available.
Our results have several implications to our understanding and modeling of Arctic climate.
• Cloud ice microphysical processes are important contributors to the Arctic low cloud amount annual cycle and
therefore are important to the seasonality of the Arctic surface energy budget and sea ice cover.
• Mean Arctic low cloud amount is strongly constrained by atmospheric variability, namely by the lower
tropospheric stability and mid-tropospheric vertical motion fields.

- • Lower tropospheric stability plays an important role in explaining the inter-model differences in low cloud
- amount.
- • Cloud microphysical parameterizations drive significant inter-model differences in Arctic cloud amount and its
- annual cycle.
- • Improved modeling of the Arctic cloud amount annual cycle, and its influences on Arctic climate variability and
- change requires observational constraints on ice microphysical processes, particularly on cloud phase partitioning
- and ice formation mechanisms.
- • The general thinking that models producing too much ice then desiccate supercooled liquid and yield fewer clouds
- does not explain model biases in low cloud amount. Our results indicate that in winter a larger ice condensate
- fraction supports larger low cloud amounts, likely because models simulate very little supercooled liquid in
- winter. Larger supercooled liquid water is associated with larger low cloud amounts in summer.
- • Lastly, we were surprised to find that models treating cloud ice and liquid condensate as separate prognostic
- variables simulate larger ice condensate fractions than those that treat total cloud condensate as a prognostic
- variable and use a temperature-dependent phase partitioning.

In closing, Arctic cloud amount plays a significant role in shaping Arctic climate system evolution. Given the
stark evidence that the Arctic climate is changing more rapidly than the rest of the globe, an improved modeling
capability in this highly varying, highly susceptible, and geopolitically important region is urgent. A better
understanding of Arctic clouds is vital to providing this improved capability. This analysis advances our understanding
of the factors that drive Arctic cloud behavior in climate models and points to unresolved issues in ice microphysics
as the likely explanation. Thus, our results underscore the vital need for observational constraints on these critical
processes.
**Code availability**: Computer code used for the analysis was written in IDL and is available from the authors upon
request.
**Author Contributions:** PCT and RCB formulated the studied, performed the analysis, and PCT, RCM, YL, and
DWJT
**Competing Interests:** The authors declare no competing interests.
**Data Availability:** The CMIP5 model data analyzed and supports the finding of this study are deposited in the Earth
System Grid Federation Peer-to-Peer enterprise system and available at https://esgf-node.llnl.gov/projects/esgf-llnl/.
**Acknowledgements:** The authors would like to thank Abhay Devasthale and an anonymous reviewer for the helpful
comments. This work is funded by the NASA Program grant number NNH16ZDA001N-NDOA. We acknowledge
the World Climate Research Programme's Working Group on Coupled Modelling, which is responsible for CMIP.
For CMIP the U.S. Department of Energy's Program for Climate Model Diagnosis and Intercomparison provides
coordinating support and leads development of software infrastructure in partnership with the Global Organization for
Earth System Science Portals.

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

**Table 1: Summary of cloud fraction and microphysical parameterization schemes for CMIP5 models.**

| Model | Institution | Cloud Fraction and Microphysics | Reference |
|---|---|---|---|
| ACCESS1.0 | Commonwealth Scientific and Industrial Research Organisation, Bureau of Meteorology | PDF-based diagnostic cloud scheme with bulk single moment microphysics | *Collier and Uhe* [2012]; *Bi et al.* [2012a] |
| ACCESS1.3 | Commonwealth Scientific and Industrial Research Organisation, Bureau of Meteorology | PDF-based prognostic cloud scheme with bulk single moment microphysics | *Collier and Uhe* [2012]; *Bi et al.* [2012a] |
| BCC-CSM1.1 | Beijing Climate Center | non-PDF prognostic cloud scheme, bulk single moment microphysics | *Wu et al.* [2008] |
| BCC-CSM1.1(m) | Beijing Climate Center | non-PDF prognostic cloud scheme, bulk single moment microphysics | *Wu et al.* [2008] |
| BNU-ESM | College of Global Change and Earth System Science, Beijing Normal University | non-PDF diagnostic cloud fraction with prognostic cloud water, bulk single moment microphysics | *Ji et al.* [2014]; *Wu et al.* [2013] |
| CanESM2 | Canadian Centre for Climate Modelling and Analysis | PDF-based diagnostic cloud scheme with bulk single moment microphysics | *Arora et al.* [2011]; *von Salzen et al.* [2013] |
| CCSM4 | National Center for Atmospheric Research | non-PDF diagnostic cloud fraction with prognostic cloud water, bulk single moment microphysics | *Gent et al.* [2011]; *Gettelman et al.* [2008] |
| CMCC-CM | Centro Euro-Mediterraneo per I Cambiamenti Climatici | PDF-based prognostic cloud scheme, double moment microphysics | *www.cmcc.it/models/cmcc-cm; Roeckner et al.* [2003] |
| CESM1-BGC | National Science Foundation, Dept. of Energy, National Center for Atmospheric Research | non-PDF diagnostic cloud fraction with prognostic cloud water, bulk single moment microphysics | *Gent et al.* [2011] |
| CESM1-CAM5 | National Science Foundation, Dept. of Energy, National Center for Atmospheric Research | Prognostic two-moment formulation of cloud liquid and ice with mass and number concentration . Multiple ice nucleation mechanisms calculated; allows for supersaturation with respect to ice | *Neale, R. et al.* [2012]; *Meehl et al.* [2013]; *Gettelman et al. 2008* |
| CNRM-CM5 | Centre National de Recherches Meteorologiques, Centre Europeen de Recherche et Formation Avancees en Calcul Scientifique | PDF-based diagnostic cloud scheme | *Voldoire et al.* [2012] |
| CSIRO-Mk3.6.0 | Commonwealth Scientific and Industrial Research Organisation in collaboration with the Queensland Climate Change Centre of Excellence | non-PDF diagnostic cloud scheme, bulk single moment microphysics | *Rotstayn et al.* [2012] |
| FGOALS-g2 | LASG, Institute of Atmospheric Physics, Chinese Academy of Sciences; and CESS, Tsinghua University | non-PDF cloud scheme, 2-moment microphysics | *Li et al.* [2013] |
| GFDL-CM3 | Geophysical Fluid Dynamics Laboratory | PDF-based prognostic cloud scheme, bulk single moment microphysics | *Donner et al.* [2011] |
| GISS-E2-H | NASA Goddard | non-PDF diagnostic cloud scheme, bulk single moment microphysics | *Menon et al.* [2010]; *Del Genio* [1996] |
| GISS-E2-R | NASA Goddard | non-PDF diagnostic cloud scheme, bulk single moment microphysics | *Menon et al.* [2010]; Del Genio [1996] |
| INM-CM4 | Institute for Numerical Mathematics | non-PDF diagnostic cloud scheme, bulk single moment microphysics | *Volodin, Diansky, and Gusev* [2010] |
| IPSL-CM5A-LR | Institut Pierre-Simon Laplace | PDF-based diagnostic cloud scheme with bulk single moment microphysics | *Dufresne et al.* [2013] |
| IPSL-CM5A-MR | Institut Pierre-Simon Laplace | PDF-based diagnostic cloud scheme with bulk single moment microphysics | *Dufresne et al.* [2013] |
| MIROC5 | Atmosphere and Ocean Research Institute (The University of Tokyo), National Institute for Environmental Studies, Japan Agency for Marine-Earth Science and Technology | PDF-based prognostic cloud scheme with bulk single moment microphysics | *Watanabe et al. [2010]* |
| MPI-ESM-MR | Max Planck Institute for Meteorology | PDF-based diagnostic cloud fraction | *Raddatz et al.* [2007] |
| MPI-ESM-LR | Max Planck Institute for Meteorology | PDF-based diagnostic cloud fraction | *Raddatz et al.* [2007] |
| MRI-CGCM3 | Meteorological Research Institute | PDF-based diagnostic cloud scheme with double moment microphysics | *Yukimoto et al.* [2011] |
| NorESM1-M | Norwegian Climate Centre | non-PDF diagnostic cloud fraction with prognostic cloud water, bulk single moment microphysics | *Kirkevag et al.* [2013]; *Rasch and Kristjansson* [1998] |
| NorESM1-ME | Norwegian Climate Centre | non-PDF diagnostic cloud fraction with prognostic cloud water, bulk single moment microphysics | *Kirkevag et al.* [2013]; *Rasch and Kristjansson* [1998] |


**Table 2: Annual mean atmospheric conditions for MERRA-2, Group 1, Group 2 for ocean and land, and the 95% confidence interval for the difference in means (Group 1 – Group 2).**

## OCEAN

| | MERRA-2 | GROUP 1 | GROUP 2 | 95% CI OF $\mu_{G1}$ - $\mu_{G2}$ |
|---|---|---|---|---|
| LTS (K) | 20.76 | 20.75 | 23.30 | $-2.55 < \mu_{G1}\text{-}\mu_{G2} < -2.54$ |
| $-\omega_{500}$ (hPa day$^{-1}$) | 1.16 | 0.90 | -0.33 | $1.21 < \mu_{G1}\text{-}\mu_{G2} < 1.24$ |
| SHF (W m$^{-2}$) | 12.33 | 4.55 | 5.69 | $-1.167 < \mu_{G1}\text{-}\mu_{G2} < -1.119$ |
| LHF (W m$^{-2}$) | 13.78 | 11.85 | 10.23 | $1.59 < \mu_{G1}\text{-}\mu_{G2} < 1.64$ |
| LOW CLOUD (%) | 24.20 | 25.60 | 22.66 | $2.938 < \mu_{G1}\text{-}\mu_{G2} < 2.96$ |
| HIGH CLOUD (%) | 16.80 | 18.00 | 12.65 | $5.35 < \mu_{G1}\text{-}\mu_{G2} < 5.36$ |
| SIC (%) | | 76.60 | 81.30 | $-4.71 < \mu_{G1}\text{-}\mu_{G2} < -4.64$ |
| LOW-LEVEL RH (%) | 84.00 | 79.50 | 85.20 | $-5.72 < \mu_{G1}\text{-}\mu_{G2} < -5.70$ |
| LOW-LEVEL T$_A$ (K) | 262.50 | 260.90 | 260.90 | $-0.008 < \mu_{G1}\text{-}\mu_{G2} < 0.0097$ |
| CLI (g kg$^{-1}$) | 0.0016 | 0.0050 | 0.0043 | $0.00074 < \mu_{G1}\text{-}\mu_{G2} < 0.00075$ |
| CLW (g kg$^{-1}$) | 0.0197 | 0.0140 | 0.0246 | $-0.0105 < \mu_{G1}\text{-}\mu_{G2} < -0.0104$ |

## LAND

| | MERRA-2 | GROUP 1 | GROUP 2 | 95% CI OF $\mu_{G1}$ - $\mu_{G2}$ |
|---|---|---|---|---|
| LTS (K) | 20.48 | 19.90 | 21.30 | $-1.315 < \mu_{G1}\text{-}\mu_{G2} < -1.29$ |
| $-\omega_{500}$ (hPa day$^{-1}$) | -2.95 | -3.73 | -0.48 | $-3.287 < \mu_{G1}\text{-}\mu_{G2} < -3.2$ |
| SHF (W m$^{-2}$) | 1.79 | 0.74 | 2.20 | $-1.48 < \mu_{G1}\text{-}\mu_{G2} < -1.425$ |
| LHF (W m$^{-2}$) | 21.10 | 15.32 | 13.50 | $1.78 < \mu_{G1}\text{-}\mu_{G2} < 1.83$ |
| LOW CLOUD (%) | 15.10 | 22.67 | 20.50 | $2.148 < \mu_{G1}\text{-}\mu_{G2} < 2.175$ |
| HIGH CLOUD (%) | 17.30 | 21.15 | 14.7 | $6.40 < \mu_{G1}\text{-}\mu_{G2} < 6.42$ |
| LOW-LEVEL RH (%) | 80.80 | 76.50 | 82.60 | $-6.12 < \mu_{G1}\text{-}\mu_{G2} < -6.09$ |
| LOW-LEVEL T$_A$ (K) | 265.30 | 263.90 | 263.60 | $0.267 < \mu_{G1}\text{-}\mu_{G2} < 0.293$ |
| CLI (g kg$^{-1}$) | 0.0008 | 0.0045 | 0.0049 | $-0.00034 < \mu_{G1}\text{-}\mu_{G2} < -0.00032$ |
| CLW (g kg$^{-1}$) | 0.0174 | 0.0160 | 0.0276 | $-0.0115 < \mu_{G1}\text{-}\mu_{G2} < -0.0114$ |

**Table 3: Summary of the average low cloud amount for each group from model output and as computed using**
**Equation (1).**

|  | GROUP 1 | GROUP 2 |
| --- | --- | --- |
| **DJF domain-averaged LCA** | 29.0% | 17.2% |
| **DJF LCA from Eq. (1)** | 29.8% | 16.3% |
| **JJA domain-averaged LCA** | 23.1% | 27.0% |
| **JJA LCA from Eq. (1)** | 21.8% | 26.1% |



**Table 4: Summary of decomposition results attributing Group 1 minus Group 2 differences in the average low cloud amount following Equation (2).**

**AVERAGE LCA CONSTRUCTED FROM [LTS, -ω$_{500}$]**

|  | $\overline{\Delta LCA}_{G1-G2}$ | $\overline{\delta LCA}_{G1-G2}$ | $\delta LCA_{G1-G2} \cdot RFO_{G1}$ | $LCA_{G1} \cdot \delta RFO_{G1-G2}$ |
|---|---|---|---|---|
| **WINTER** | 11.80% | 13.30% | 13.10% | 0.17% |
| **SUMMER** | -3.84% | -4.45% | -4.49% | 0.05% |

**AVERAGE LCA CONSTRUCTED FROM [T$_a$, RH]**

|  | $\overline{\Delta LCA}_{G1-G2}$ | $\overline{\delta LCA}_{G1-G2}$ | $\delta LCA_{G1-G2} \cdot RFO_{G1}$ | $LCA_{G1} \cdot \delta RFO_{G1-G2}$ |
|---|---|---|---|---|
| **WINTER** | 11.60% | 10.40% | 12.20% | -1.80% |
| **SUMMER** | -4.20% | -4.68% | -1.37% | -3.31% |

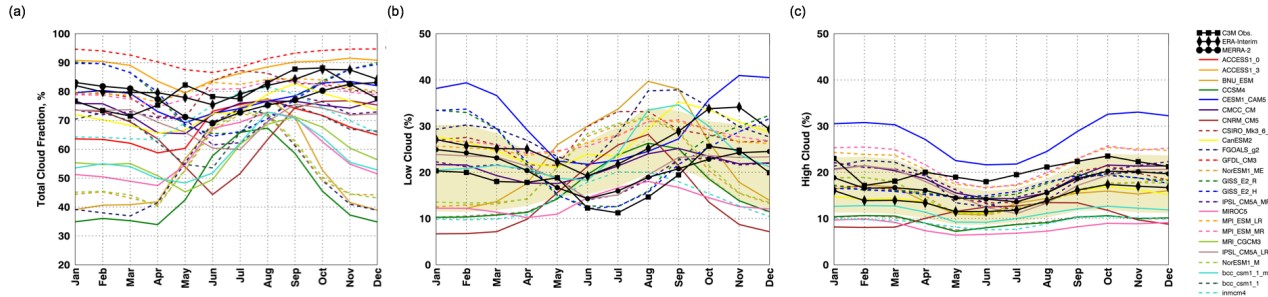

652

**Figure 1: Annual cycle of (a) total cloud amount, (b) low cloud amount (defined as cloud between 1000 – 850 hPa) and (c) high cloud amount (cloud between 500 – 300 hPa). Color lines represent individual CMIP5 models, black lines with symbols represent C3M observations and reanalysis. The yellow shading in (b)-(c) represents the ensemble mean +/- one standard deviation.**





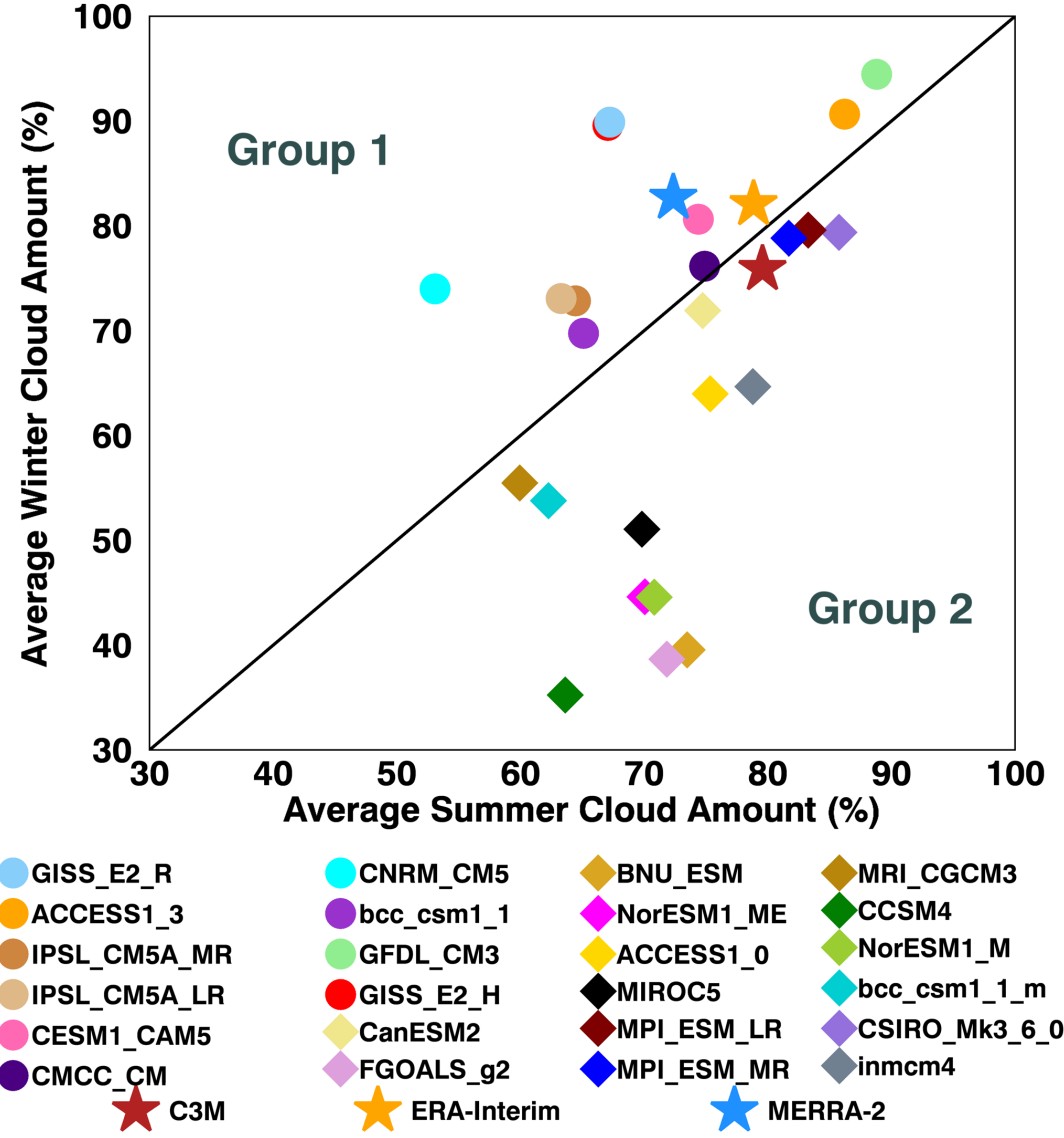

**Figure 2: Average total cloud amount in winter (DJF) vs average in summer (JJA). Models above the 1:1 line (maximum cloud amount in winter; circle symbols) are defined as Group 1 and those below the 1:1 line (maximum cloud amount in summer; square symbols) are Group 2. The star symbols represent C3M observations (red), ERA-Interim (orange), and MERRA-2 (blue).**

666

667

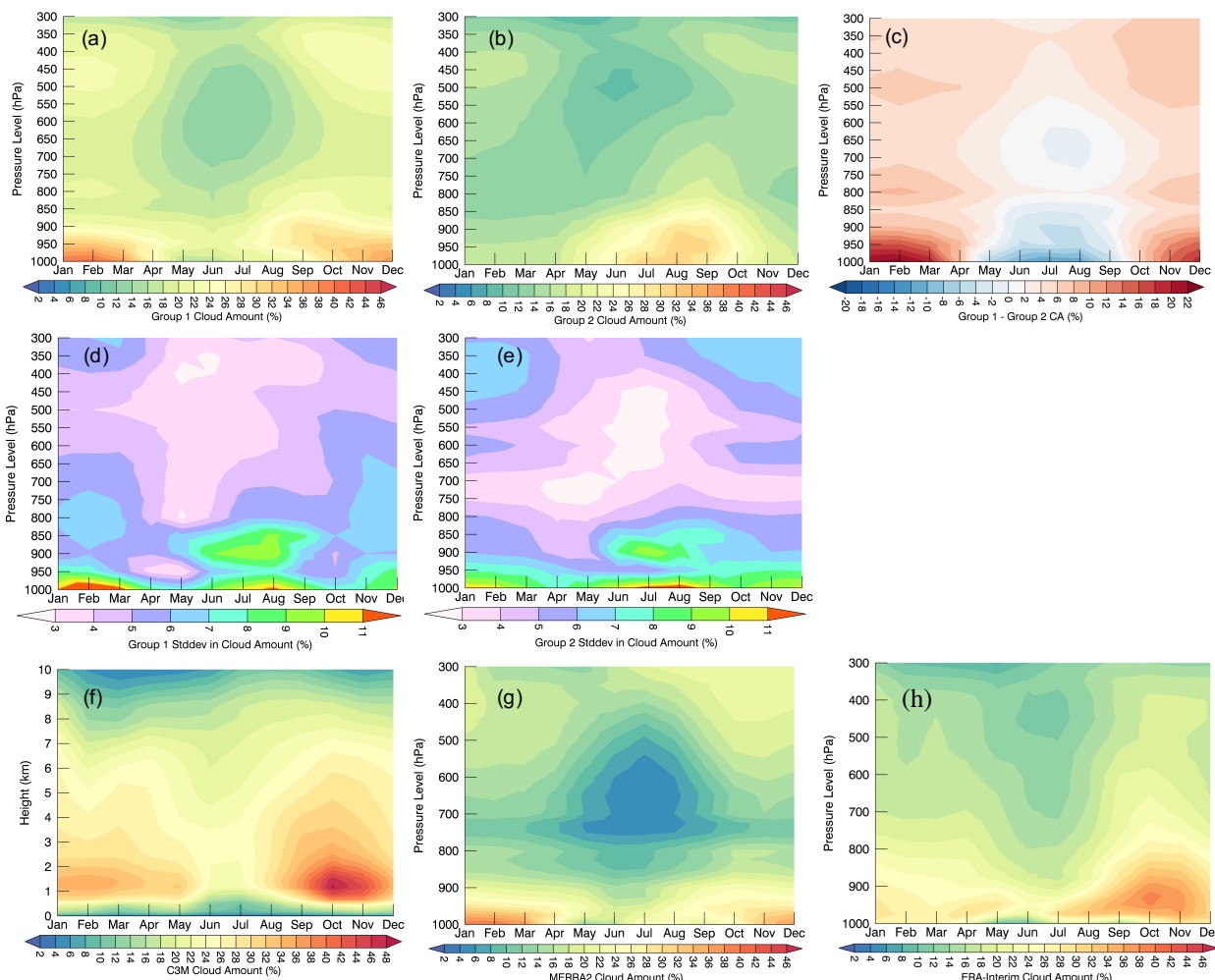

**Figure 3: Vertically-resolved mean cloud amount annual cycle for (a) Group 1, (b) Group 2, and (c) Group 1 – Group 2 and the vertically resolved standard deviation across (d) Group 1 and (e) Group 2 members are shown. Observational profiles of cloud amount are shown for (f) C3M, (g) MERRA-2, and (h) ERA-Interim**

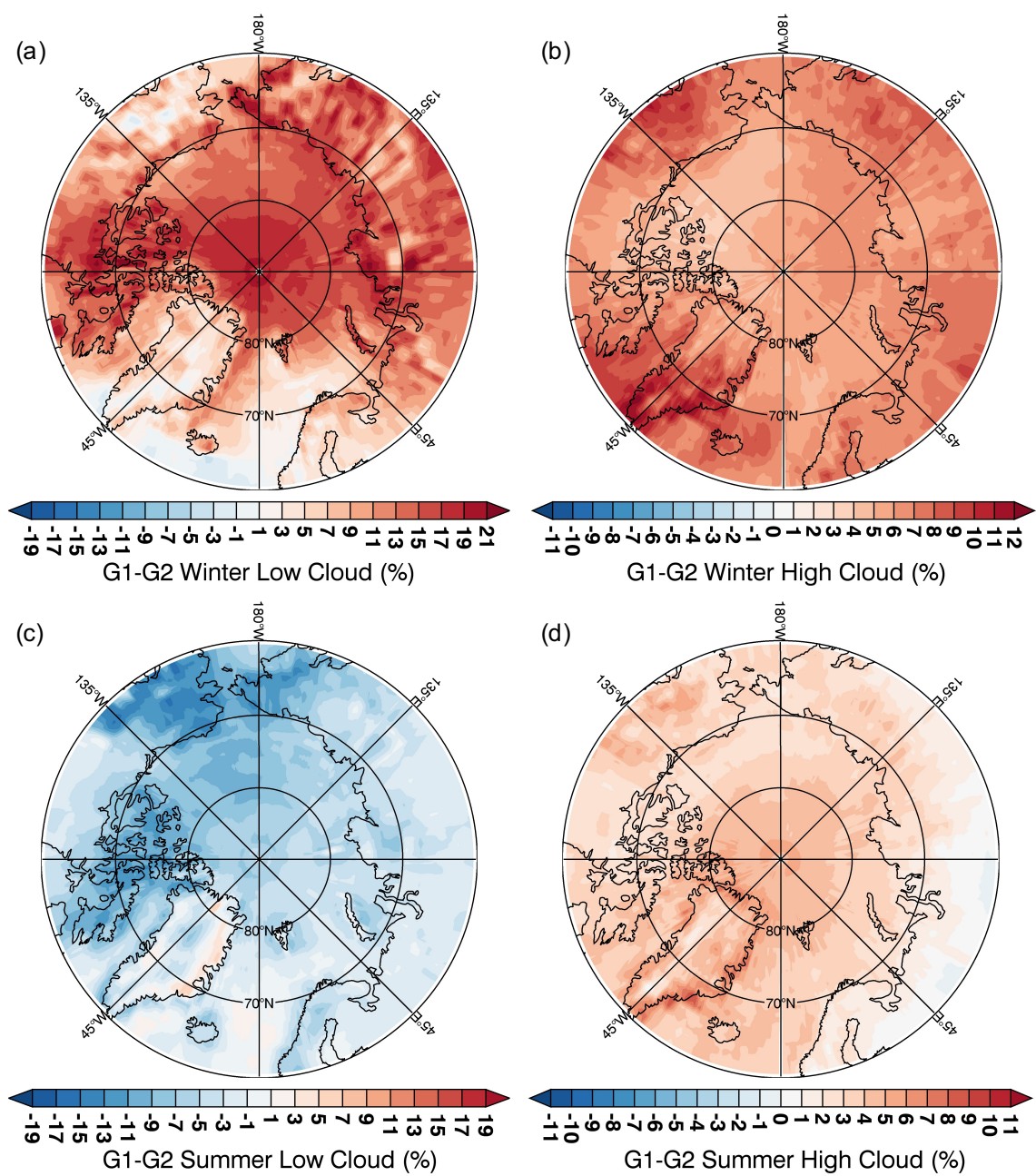


**Figure 4: Spatial variations in Group 1 minus Group 2 cloud amount differences for (a) winter low clouds, (b) winter high clouds, (c) summer low clouds, and (d) summer high clouds.**



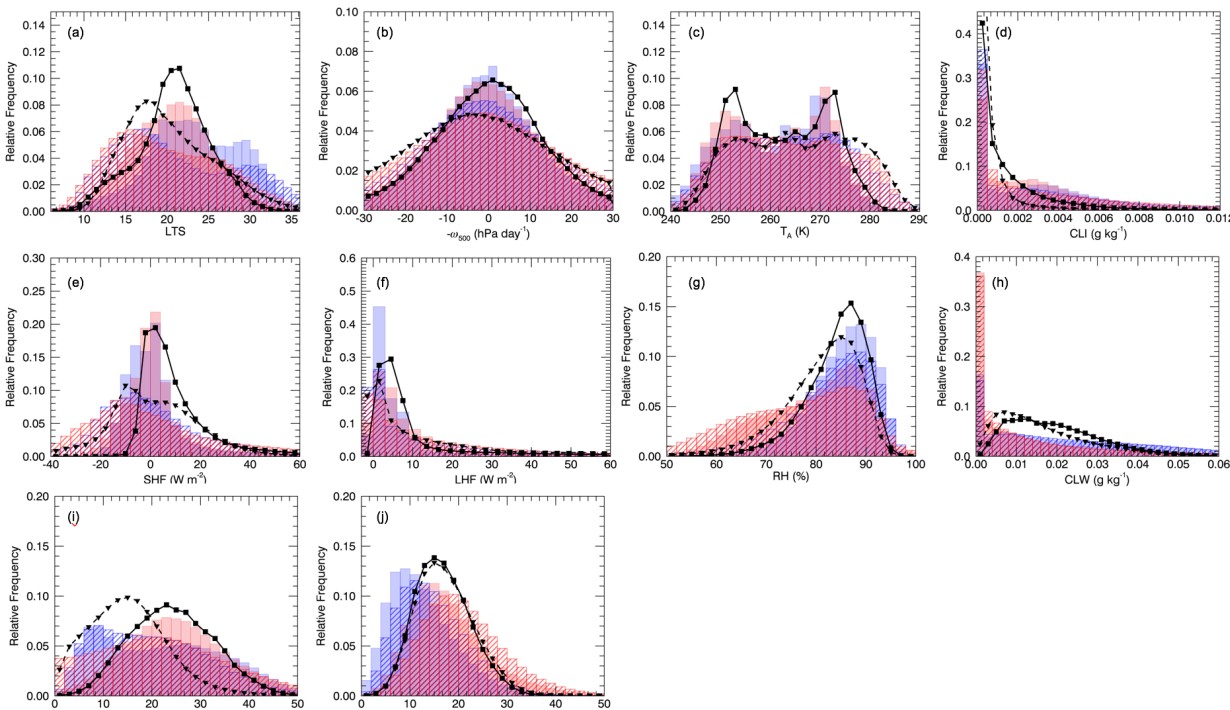


**Figure 5: Probability distributions of (a) LTS, (b) -$\omega_{500}$, (c) low-level T$_A$, (d) CLI, (e) SHF, (f) LHF, (g) RH, (h)**
**CLW, (i) low cloud amount, and (j) high cloud amount. Red shading denotes Group 1, blue denotes Group 2,**
**solid fill represents ocean grid boxes, and cross-hatching represents land grid boxes. The solid black line shows**
**MERRA-2 reanalysis values for ocean (square symbol) and land (triangle symbol). Distributions include all**
**months of the year.**

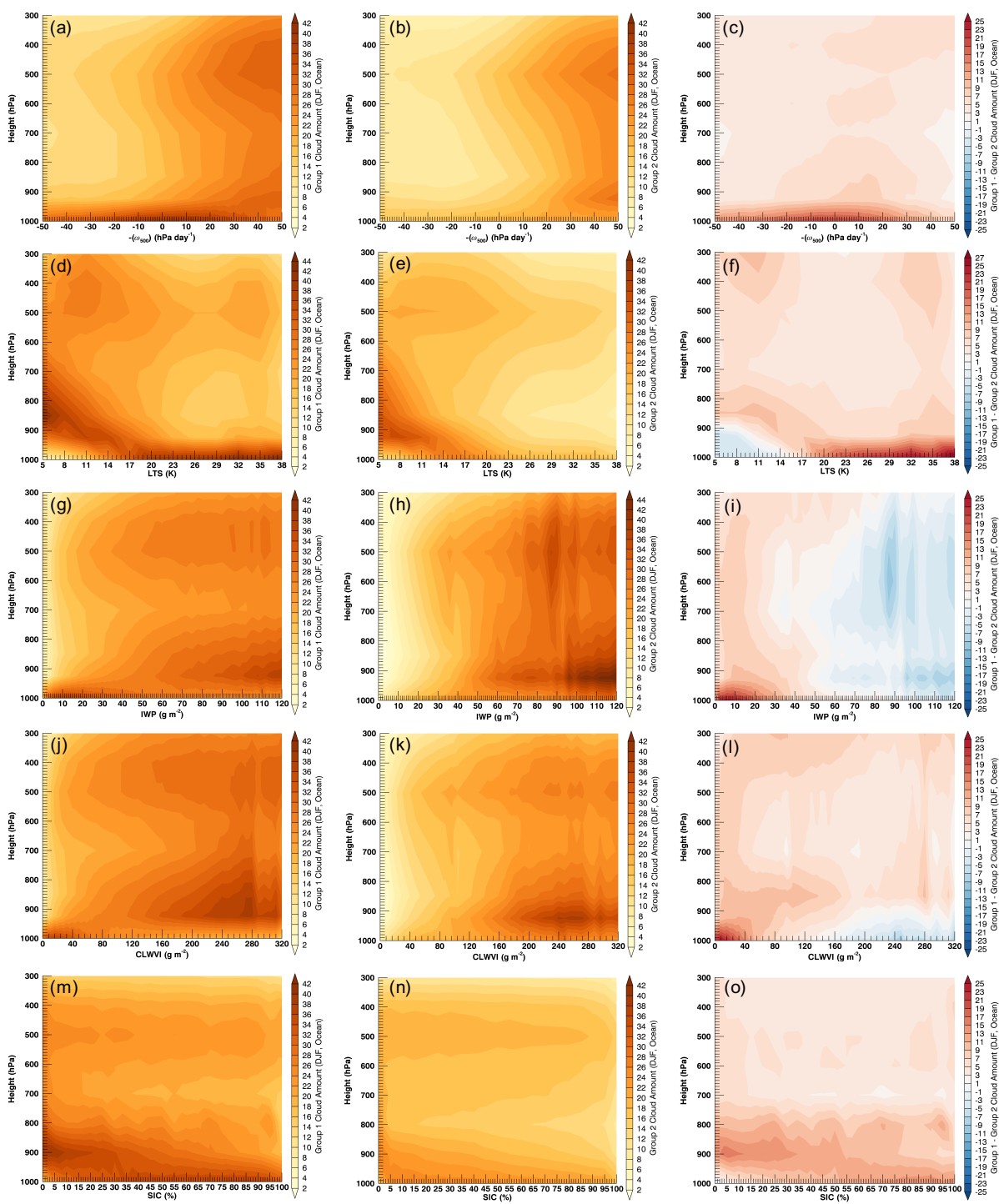

**Figure 6: Vertically-resolved, DJF average cloud amount stratified by -ω$_{500}$ for (a) Group 1, (b) Group 2, and (c) Group 1 minus Group 2, LTS for (d) Group 1, (e) Group 2, and (f) Group 1 minus Group 2, IWP for (g) Group 1, (h) Group 2, and (i) Group 1 minus Group 2, CLWVI for (j) Group 1, (k) Group 2, and (l) Group 1 minus Group 2, and SIC for (m) Group 1, (n) Group 2, and (o) Group 1 minus Group 2. All panels are for ocean.**

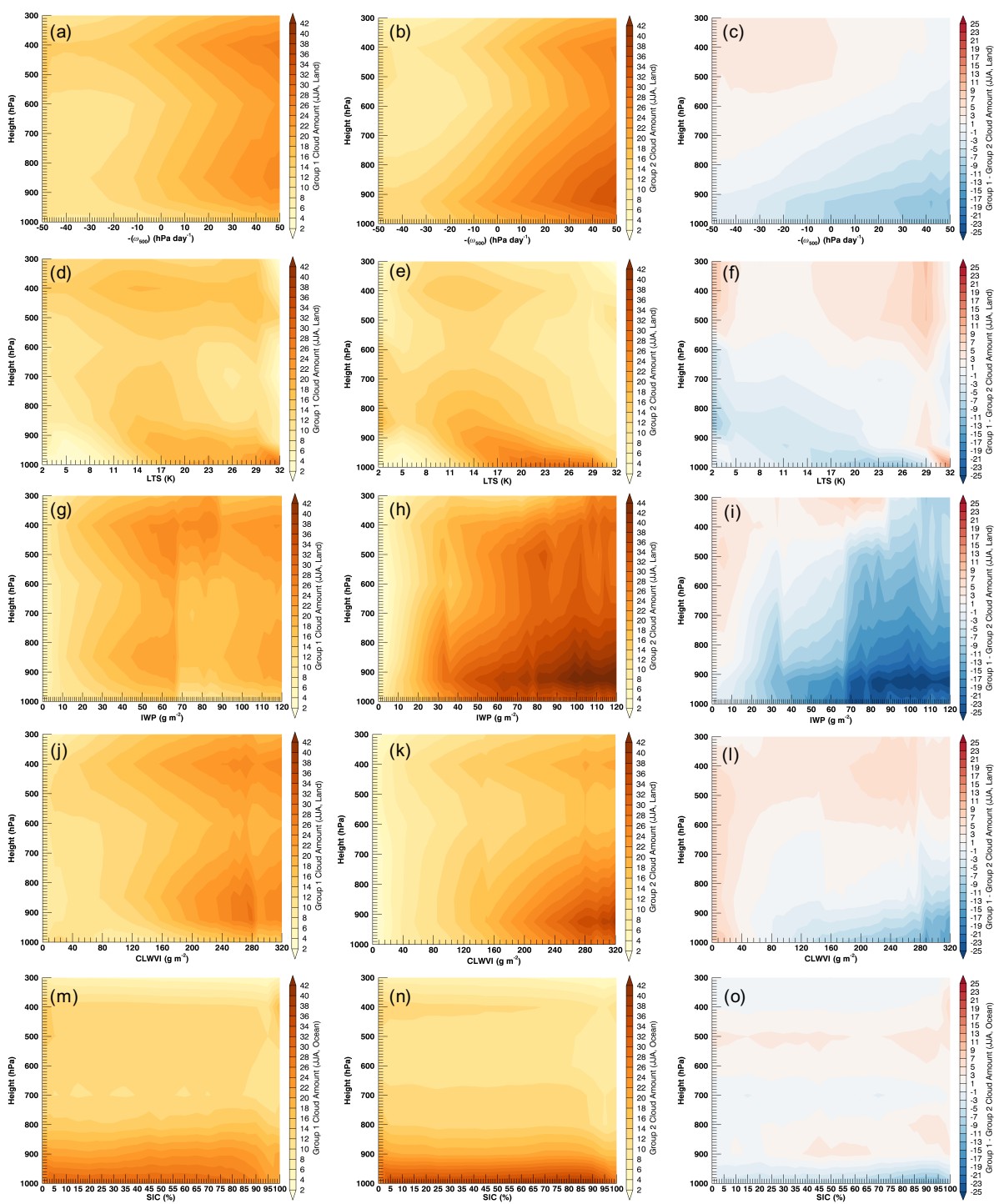


**Figure 7: Vertically-resolved, JJA cloud amount stratified by -ω$_{500}$ for (a) Group 1, (b) Group 2, and (c) Group**

**1 minus Group 2, LTS for (d) Group 1, (e) Group 2, and (f) Group 1 minus Group 2, IWP for (g) Group 1, (h)**
**Group 2, and (i) Group 1 minus Group 2, CLWVI for (j) Group 1, (k) Group 2, and (l) Group 1 minus Group**
**2, and SIC for (m) Group 1, (n) Group 2, and (o) Group 1 minus Group 2. All panels are over land except for**
**SIC.**

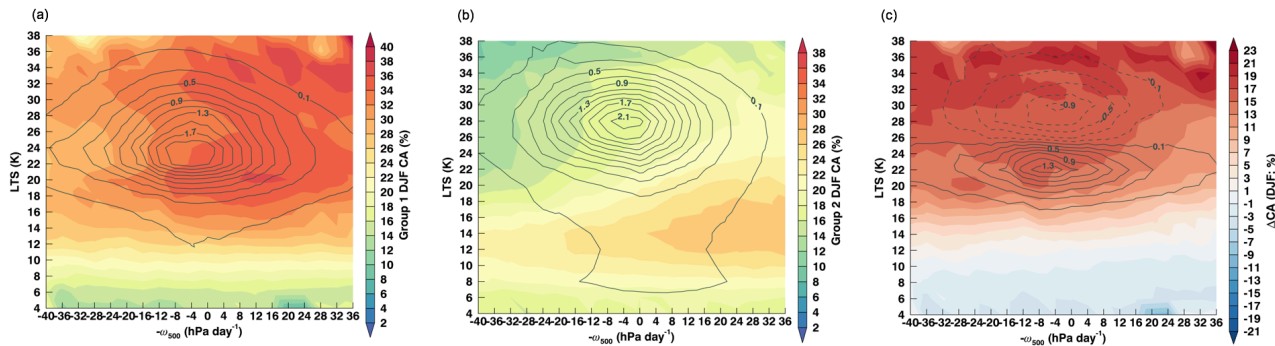


**Figure 8: Contours of average low cloud amount for DJF in the LTS and -ω$_{500}$ joint distribution for (a) Group**
**1, (b) Group 2, and (c) Group 1 minus Group 2. The frequency of occurrence each LTS and -ω$_{500}$ bin is contoured**
**in solid black with an interval of 0.2%.**


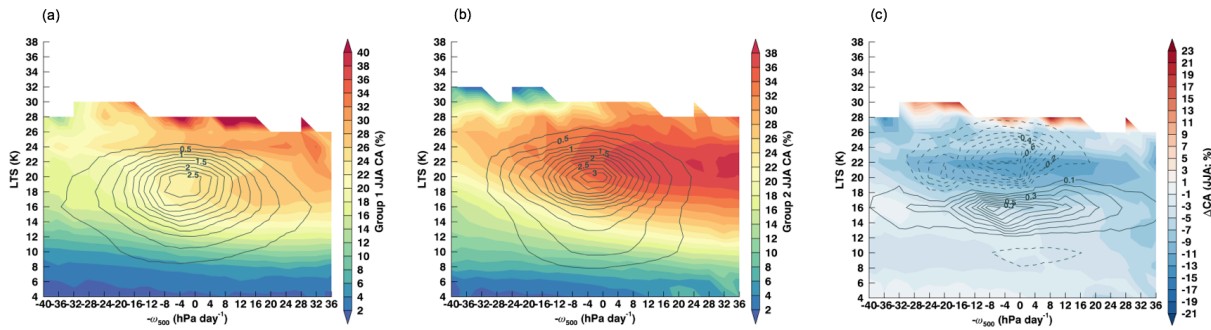


**Figure 9: Contours of average low cloud amount for JJA in the LTS and -ω$_{500}$ joint distribution for (a) Group**
**1, (b) Group 2, and (c) Group 1 minus Group 2. The frequency of occurrence each LTS and -ω$_{500}$ bin is contoured**
**in solid black with an interval of 0.2%.**


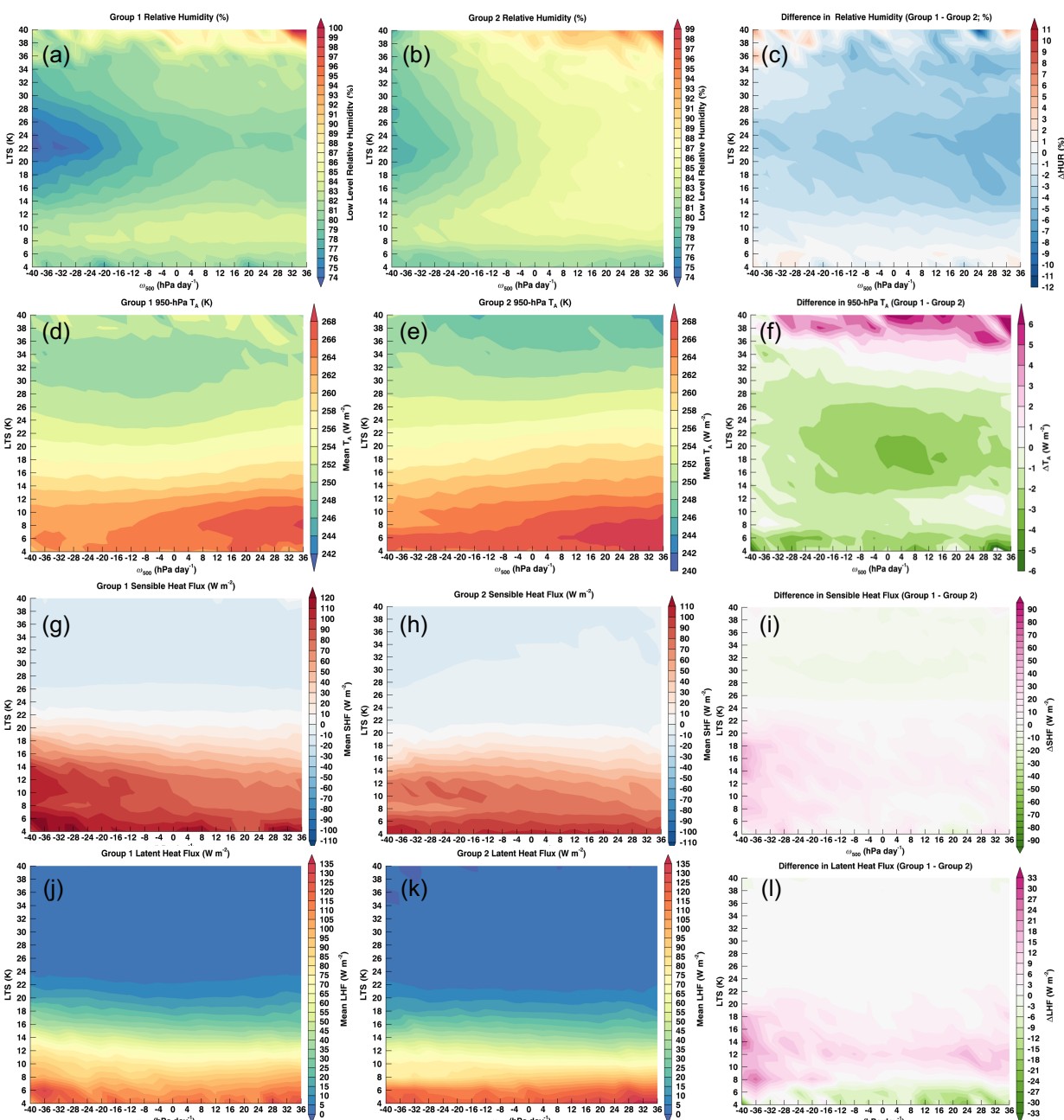

Figure 10: Contours of DJF atmospheric and surface conditions in the LTS and -$\omega_{500}$ joint distribution for (left column) Group 1, (middle column) Group 2, and (right column) Group 1 minus Group 2 for (a-c) RH, (d-f) $T_A$ at 950hPa, (g-l) SHF, and (j-l) LHF.


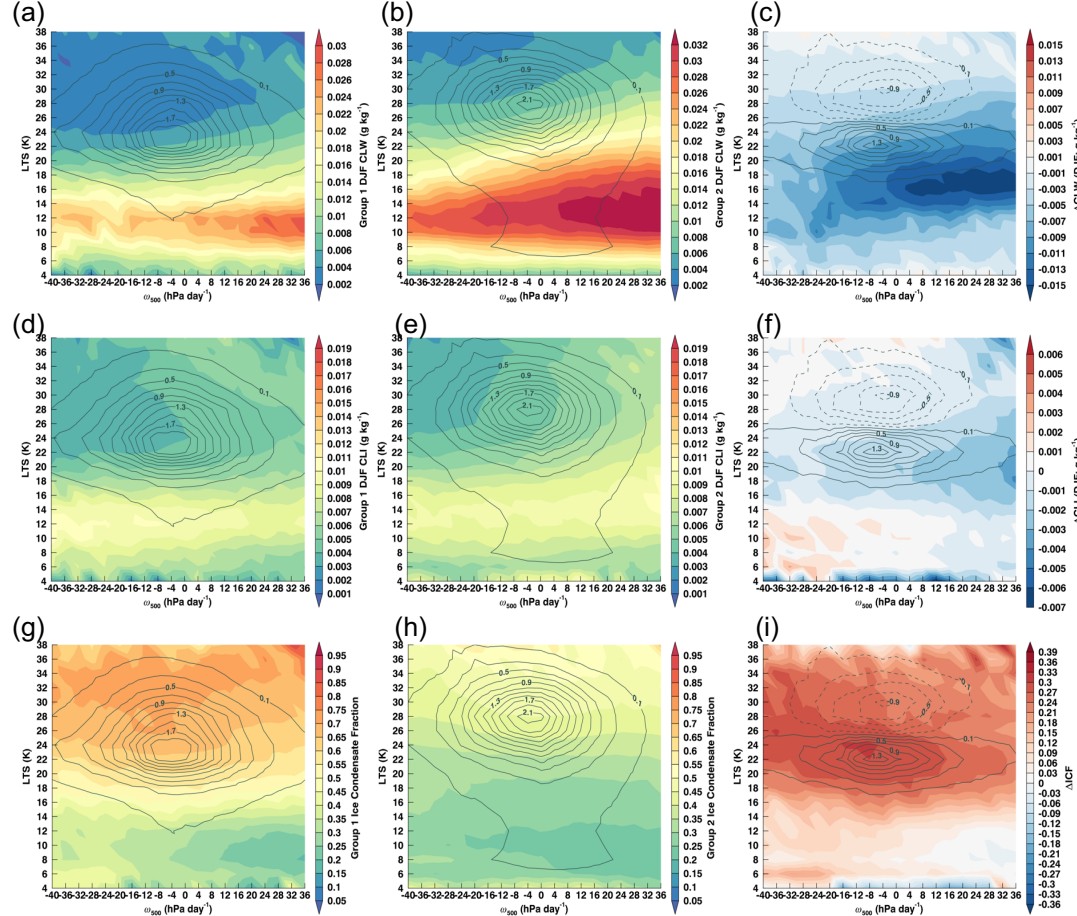

**Figure 11: Contours of winter low cloud CLW for (a) Group 1, (b) Group 2, and (c) Group 1 minus Group 2,**
**CLI (d) Group 1, (e) Group 2, and (f) Group 1 minus Group 2, and ice condensate fraction for (g) Group 1, (h)**
**Group 2, and (i) Group 1 minus Group 2.**



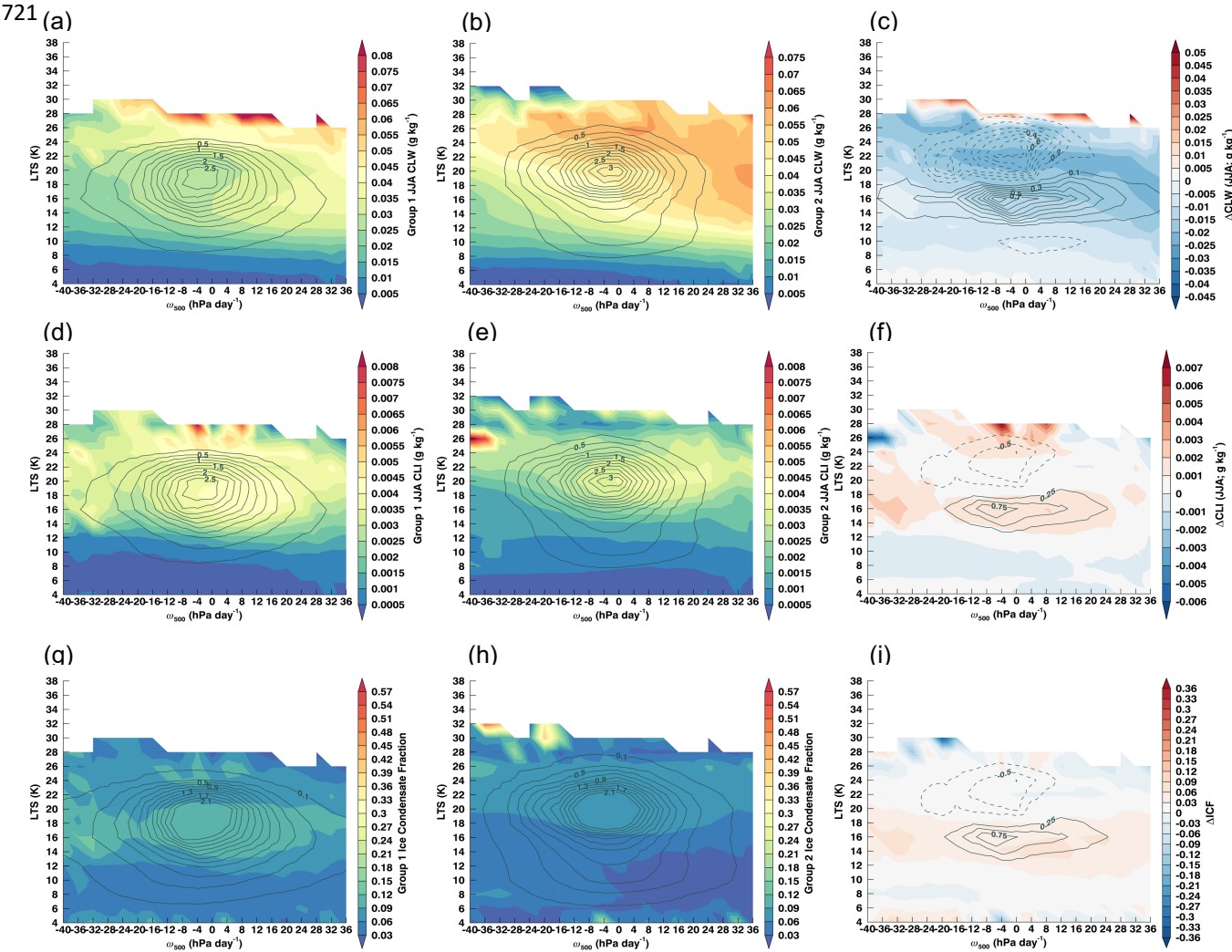

**Figure 12: Contours of JJA low cloud CLW for (a) Group 1, (b) Group 2, and (c) Group 1 minus Group 2, CLI**
**(d) Group 1, (e) Group 2, and (f) Group 1 minus Group 2, and ice condensate fraction for (g) Group 1, (h)**
**Group 2, and (i) Group 1 minus Group 2.**

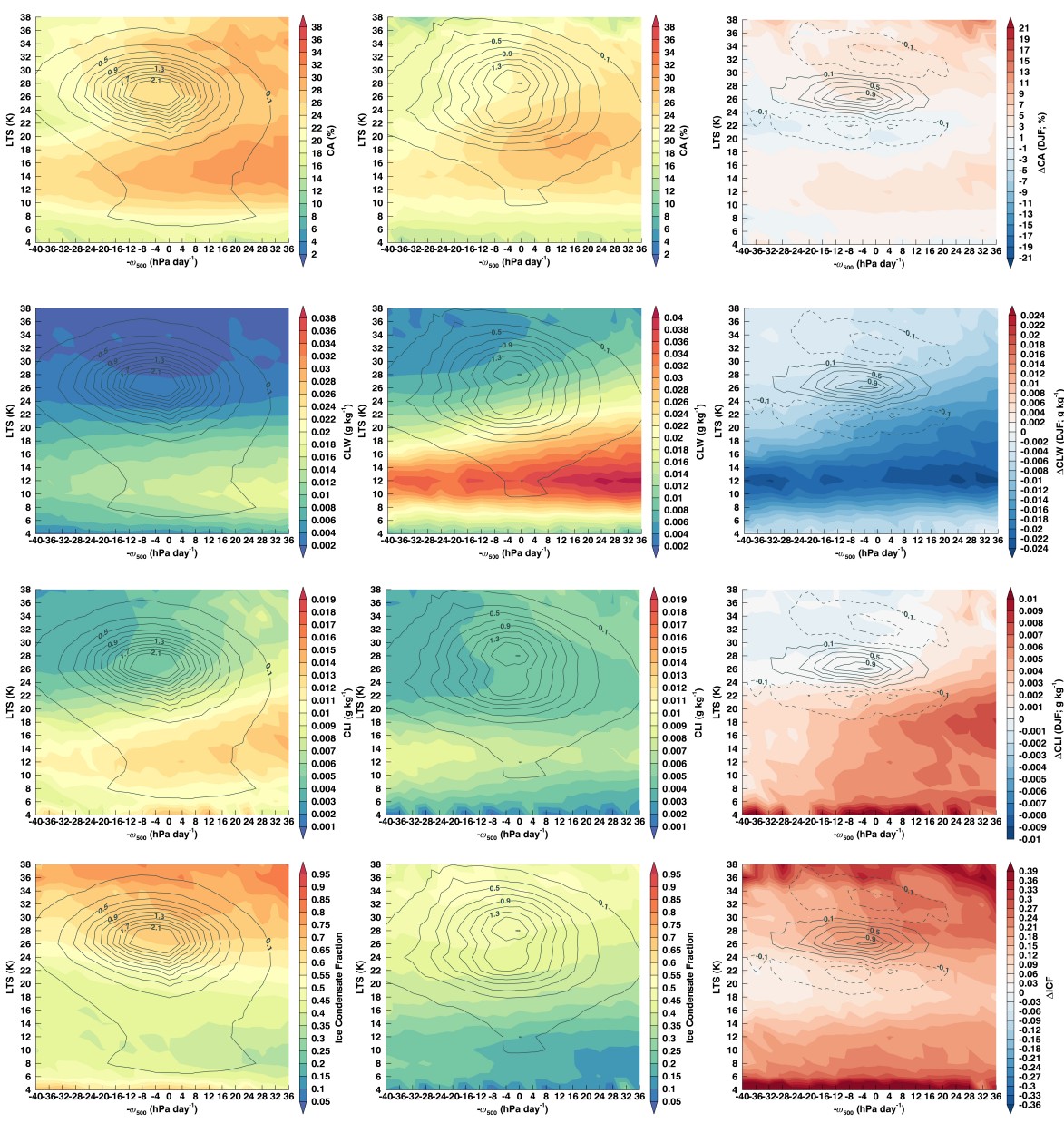


**Figure 13: Contours of winter low cloud amount for (a) Group A, (b) Group B, and (c) Group A minus Group B, low liquid water mixing ratio (d) Group A, (e) Group B, and (f) Group A minus Group B, low cloud ice water mixing ratio Group A (g), Group B (h), and Group A minus Group B (i). and ice condensate fraction is shown in the bottom panels for Group A (j), Group B (k), and (l) Group A minus Group B.**

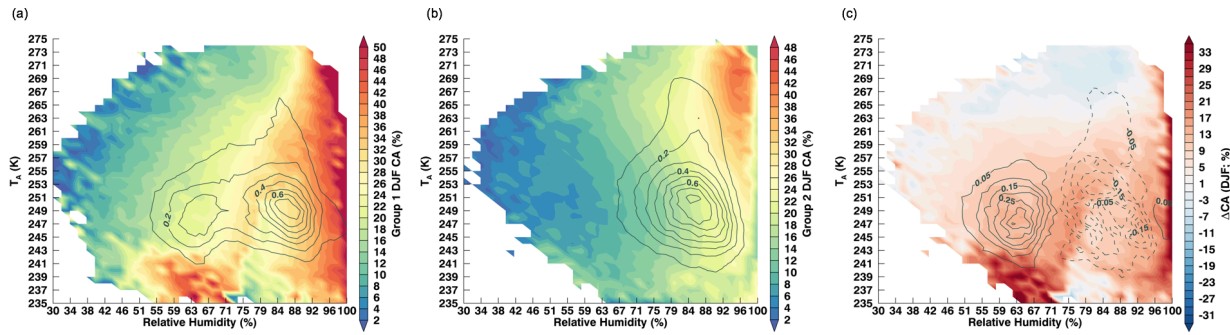


**Figure 14: Contours of average low cloud amount for DJF in the $T_A$-RH joint distribution for (a) Group 1, (b)**
**Group 2, and (c) Group 1 minus Group 2. The frequency of occurrence of $T_A$-RH bins is contoured in solid**
**black with an interval of 0.2%.**


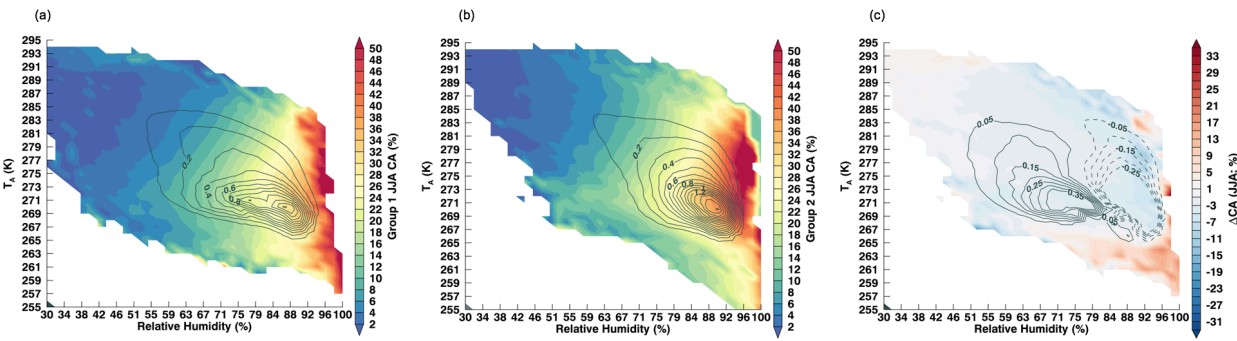


Figure 15: Contours of average low cloud amount for JJA in the $T_A$-RH joint distribution for (a) Group 1, (b) Group 2, and (c) Group 1 minus Group 2. The frequency of occurrence of $T_A$-RH bins is contoured in solid black with an interval of 0.2%.