# Peer review of "Arctic cloud annual cycle biases in climate models"

_Atmospheric Chemistry and Physics, 2018_

## Referee Comment (RC1) · Anonymous Referee #1 · 12 Jan 2019

This manuscript analyzes climate model simulations of Arctic clouds, to understand the seasonal cycle and what drives it. It provides some interesting analysis, by grouping models by their seasonal cycle. The paper may be suitable for publication in ACP subject to major revisions.

I have a few general concerns:

1. The authors need to show that the results are robust to changes in model groups. Perhaps 1/3 of the models are very close to a 1:1 line they use to select models. What happens if you change the grouping of models? Does it change the results?

2. The authors claim that since their results agree with earlier work, it is fine to use

monthly data. That is not sufficient. They are averaging over regimes that may yield very different results, and they need to verify with a single model perhaps that monthly data for joint PDFs for example matches high frequency (daily or higher) data.

3. The lack of ice fraction is limiting. Analysis shows ice and liquid, with no sense of what the fraction of ice is. This is related to #2 above.

4. The authors need to document models better. There needs to be a table of models with references.

5. In addition, it would be particularly useful to group those models which have ice supersaturation and look at their results.

6. There is minimal use of observations and comparison with observations in this work. It is hard to tell what is right, would like to see more comparisons against observations, and discussion and conclusions which focus on comparisons with observations. Which group isomer like observations?

Specific comments:

Page 1, L16: model group is a strange term.

Page 1, L18: thermodynamic characteristics is ambiguous.

Page 1, L19: I assume more means Higher fraction?

Page 1, L21: can you separate ice nucleation mechanisms or are you speculating? Most models do not do anything for ice nucleation and just have partition.

Page 1, L23: can you say anything about the bias and what the ice partitioning should be?

Page 2, L57: do you include/account for this bias in your analysis?

Page 2, L72: what does ice formation do to cloud amount? It's implied here but this might need another sentence of explanation.

Page 3, L77: but observations may be uncertain as well. Probably by more than 15%.

Page 3, L103: can you actually determine mechanisms from monthly mean data?

Page 4, L112: use of monthly means does not sound like an improvement.

Page 4, L113: I am skeptical of a monthly mean joint distribution. How do you deal with averaging over different regimes in a month?

Page 4, L119: you have not described the models. Where does that happen?

Page 4, L124: there are maybe 10 models that are very close to the 1:1 line. How do results change if you shift the cloud amount by +/-5% and redistribute models.

Page 4, L125: the C3M Observations are quite close to the 1:1 line. Is there really a defined maximum in winter? You might need to be more careful on the classification here, since the annual cycle may not be so much winter-summer as peaking in fall.

Page 4, L130: I guess one concern I have is that you have not addressed what a cloud is, and what a threshold water content is. How are you sure you are comparing apples with apples between the models and observations?

Page 4, L132: before a results section, you need to describe the models in a paragraph, along with a table listing the models and appropriate reference for each one.

Page 5, L153: why neglect Fall? The peak low cloud amount is in fall, not in winter.

Page 5, L164: I'm not sure I would say that the low cloud differences are spatially uniform. Differences seem lower over open water than sea ice for example, and largest differences are over land.

Page 6, L191: shouldn't you do this by season (winter-summer) or at least comment on differences between winter and summer PDFs. Maybe show a sub set?

Page 6, L198: since these are monthly means, how do you account for averaging across regimes. How do you know you are really seeing cloud influencing factors?

Page 6, L215: why are there vertical stripes here? Is this one model? Does it represent anything physical?

Page 7, L238: what does 'thin' mean?

Page 8, L277: do you think you could do this as well with sub monthly data? That might be a check that you are not a averaging over meteorological regimes (I.e. Morrison et al 2011) within a month.

Page 9, L307: what do you mean by cloud microphysics here? Besides liquid and ice, you don't have any information on microphysics (size, number, precip, optical depth, etc).

Page 9, L313: but the formation of ice would lead to more rapid depletion of cloud condensate than liquid. Rather than monthly amount, you might need fraction of Ice present, but you cannot get that from monthly means I think.

Page 9, L315: which models? Is there something systematic in models that allow supersaturation over ice? This would be an important statement.

Page 9, L318: what are the summer temperatures? Are they greater than freezing? Might just be a Clausius clayperon effect of more water at higher temperatures, not an ice formation issue.

Page 9, L321: I'm not sure you have justified this argument. What shows that? There are also significant shifts in LTS in the models.

Page 10, L346: I think the statement that monthly means are okay needs more analysis. Just because the answer looks similar in a different analysis is not sufficient.

Page 10, L356: I'm not sure that makes sense. The averaging across regimes might be a problem here. Also, amount of ice may not be as important as fraction of ice.

Page 11, L379: what models in your list support ice supersaturation?

[Figure]

Page 12, L435: but you have not commented on liquid production and availability, and ice fraction. That is the key issue.

---

## Referee Comment (RC2) · Abhay Devasthale (Referee) · 25 Feb 2019

Taylor et al investigate annual cycles of cloudiness in CMIP5 models and investigate biases among them using MERRA 2 and C3M data. I believe this is a relevant topic that, given its scientific implications, needs to be investigated. I find the analysis approach, i.e. doing investigations using stratifications by various relevant surface and atmospheric parameters, informative and useful.

I also had an opportunity to read the comments posted by the other reviewer and I must say that I broadly agree with the concerns that the reviewer raises and I do hope that the authors address them adequately.

In addition, I would like to underline few points here, where further clarifica-

tions/comments are needed before the manuscript is accepted for the final publication.

1) Let us remind ourselves that we are in the Arctic, the region that has been chronically problematic not only for models, but also for observations and reanalysis datasets. I can't help but wonder if the conclusions would change if the authors use ERA-Interim/ERA5/JMA etc instead of MERRA 2. Hinging their conclusions drawn from the stratification analysis (esp LTS, w) only on MERRA 2 is a bit risky.

2) The parameters like LWP and IWP have the largest uncertainties, no matter if you analyse reanalysis or observational data. How does this play a role? Also, can all models explicitly resolve cloud ice and cloud liquid water separately? Or does the partitioning depend on the temperature profile?

3) Over the Arctic Ocean, what kind of biases in the annual cycles of cloudiness models show if they are stratified according to sea-ice conditions, for example, permanently sea-ice covered regions versus completely ice-free regions?

4) The differences in the representation of dynamical meteorology among models are also importing while interpreting the results. For example, do models show similar heat and moisture transport into the Arctic, which has a strong influence on cloudiness?

---

## Author Comment (AC1) · 6 Apr 2019

Dear Reviewer #1 and Abhay,

We would first like to thank you for your time and effort. Your insightful comments have significantly improved this paper. The clarity and substance of the manuscript have been developed, and I have included all recommended changes by the reviewers. The most significant changes made to the manuscript were the addition of Table summarizing specifics of the microphysics schemes associated with all the model used in this study and the addition of Fig. 13. Figure 13 applies an additional stratification of the results based upon the specifics of the cloud microphysics schemes: namely, 1) models that calculate both cloud ice and cloud water as prognostic cloud variables and 2)

[Figure]

models that calculate a single mixing ratio of total water and use a temperature dependent partition to determine phase. The result shows indicate that models that treat both cloud ice and cloud water as prognostic variables produce more ice and more winter low clouds than models that treat total cloud water as the prognostic variable and use a temperature dependent partition. This result directly implicates the cloud microphysical scheme differences as a key driver of the inter-model differences in the simulation of the Arctic low cloud annual cycle.

Thank you for your comments as they have strengthened the conclusions of this study.

Sincerely

Patrick C. Taylor Research Scientist NASA Langley Research Center Climate Science Branch

Attached PDF contains the full response with figures.

Reviewer Responses for Referee #1

1. The authors need to show that the results are robust to changes in model groups. Perhaps 1/3 of the models are very close to a 1:1 line they use to select models. What happens if you change the grouping of models? Does it change the results?

Thanks. We have tested that slight changes in the model grouping has a small effect on the results and does not affect our main conclusions. To do so, we created a third group, containing five models closest to the 1:1 line (hereafter Group 3; Figure 1) and constructed joint distributions of cloud amount (CA) for this group (these models are bcc-csm1-1, CMCC-CM, CanESM2, MPI-ESM-MR, and MPI-ESM-LR; 2 Group 1 models and 3 Group 2 models). These models have smaller differences between average winter and summer CA compared to other models in their respective groups, thus we wouldn't expect the joint distributions for this group to resemble either Group 1 or Group 2 explicitly. Below is joint distribution for DJF for Group 3, to be compared to Fig. 8 in the paper. The table on the right shows average DJF CA for the ensemble,

[Figure]

Group 1, Group 2, and Group 3. Ensemble Mean Group 1 Group 2 Group 3 Avg. DJF CA (%) 21.9% 28.3% 18.6% 25.9%

The joint distribution for Group 3 contains features present in the joint distributions of both Group 1 and 2 as expected, given that Group 3 is made up of models from each group. For DJF, CA increases with increasing -ïĄů500 for low-medium stability (similar to Group 2) but with larger average cloud amount (similar to Group 1). Also similar to Group 1 is the larger CA present at high stability and rising motion. The average values of CA from the table indicate that Group 1 CA > Group 3 CA > Group 2 CA, as expected for Group 3 given that it contains models from both Group 1 and 2. These examples are given to show that the 1:1 line separating Groups 1 and 2 is a good measure for group selection- if this were not the case then we might expect Group 3 joint distributions to resemble either Group 1 or Group 2. Since Group 3 joint distributions show features from both groups, this is an indication that even the models closest to the 1:1 still represent the low cloud responses of their respective groups.

2. The authors claim that since their results agree with earlier work, it is fine to use monthly data. That is not sufficient. They are averaging over regimes that may yield very different results, and they need to verify with a single model perhaps that monthly data for joint PDFs for example matches high frequency (daily or higher) data.

To address this concern, joint distributions of low cloud amount binned by LTS and -ïĄů500 were constructed using daily data (Figure 2)from IPSL-CM5A-LR for winter (DJF) months and summer (JJA) months for the historical period 1979-2005. Additionally, eqn (1) from the paper was also calculated using the daily data to confirm the validity of monthly data. The CMIP5 archive only had daily vertical cloud amount available for one model, (IPSL-CM5A-LR, the one from Group 1). The results from this model are presented below:

Equation 1: (LCA) $\hat{\text{E}}=\sum\_(i,j)LCA(LTS\_i,-\omega$ãĂŮ$\_{(500,j)}$ )*RFO(ãĂŰLTSãĂŮ_i,ãĂŰ-$\omega$ãĂŮ$\_{(500,j)}$)ãĂŮ, describing the weighted sum of low

cloud amount over LTS and -ïĄů500 from each i,j bin where LCA(ãĂŰLTSãĂŮ_i,ãĂŰ-ωãĂŮ_(500,j)) ) is the low cloud amount as a function of LTS and -ïĄů500 and RFO(ãĂŰLTSãĂŮ_i,ãĂŰ-ωãĂŮ_(500,j)) is the relative frequency of occurrence of each LTS and -ïĄů500 bin. Applying (1) to daily data from IPSL-CM5A-LR reproduces the domain-averaged LCA with the same accuracy as shown by monthly data in Table 2.

IPSL-CM5A-LR DJF domain-averaged LCA: 25.95% IPSL-CM5A-LR DJF LCA from Eq. (1): 25.91% IPSL-CM5A-LR JJA domain-averaged LCA: 16.6% IPSL-CM5A-LR JJA LCA from Eq. (1): 16.5%

Joint distributions for DJF (left) and JJA (right) low cloud amount binned by LTS and -ïĄů500 are shown below for IPSL-CM5A-LR constructed from daily data (top row) and monthly data (bottom row). Joint distributions for JJA look very similar between the daily and monthly versions: both show a strong gradient in LCA when LTS increases, and the largest LCA for high stability and rising motion. Additionally, the frequency of occurrence of LTS/-ïĄů500 regimes is similar when using either daily or monthly data. One difference between the JJA joint distributions is the presence of highly-stable regimes captured in the daily data (LTS > 25) that are not present in the monthly. However, these highly-stable regimes occur very infrequently (less than the 0.1% frequency of occurrence contour). Differences in DJF joint distributions are larger than for JJA. For low stability (LTS < 12), both daily and monthly distributions show LCA dependent on LTS with little dependence on -ïĄů500. For medium stability (12 < LTS < 26), both show similar amounts of low cloud (particularly in the most frequent regimes) but the daily data shows a slight gradient of larger LCA with increasing -ïĄů500 (this matches with the monthly joint distribution for Group 1 (Fig. 8a in the paper)). The largest differences between daily and monthly data occur for very high stability, as was the case for JJA. Daily joint distributions show the largest LCA for LTS > 34, particularly with rising motion. This is an infrequent regime, however, that the monthly distribution does not capture.

Overall, we admit that there are some shortcomings that come with using monthly data, namely the reduced dynamic range. However, we think that the use of monthly data still provides useful results especially in regimes with most frequent LTS/-ïĄů500 bins, which is also the most frequent regimes with daily data. The differences between daily and monthly data do exist, but only occur in the least frequent LTS/-ïĄů500 bins. Lastly, the vertical profiles of cloud data on daily time scale are not available in many model outputs. New text was added to the paper (line 162-167) to point out the potential differences between using daily and monthly model output.

3. The lack of ice fraction is limiting. Analysis shows ice and liquid, with no sense of what the fraction of ice is. This is related to #2 above.

This is a very helpful suggestion. The production of cloud liquid vs ice is tied to low cloud amount differences, so we have added analysis to the paper and included joint distributions of ice condensate fraction (cloud ice water mixing ratio divided by total cloud condensate mixing ratio) stratified by Ta and RH and LTS and -ïĄů500 (Figs. 11 and 12). Further, an interesting result of this discussed in the paper and below (also see response to comment #5) is that models with a temperature-dependent phase partitioning as opposed to treating cloud ice and liquid as prognostic variables simulate less cloud ice fraction.

4. The authors need to document models better. There needs to be a table of models with references.

A table of CMIP5 models (Table 1) and corresponding references has been added, along with a column containing relevant cloud fraction and microphysics schemes used for each model.

5. In addition, it would be particularly useful to group those models which have ice supersaturation and look at their results.

When adding the table showing the relevant microphysics parameterizations for each

model, as suggested by comment #4, we did not find documentation of whether or not ice supersaturation is allowed for many of the models (though from our reading, most models do not account for ice supersaturation). We did, however, find a recently published paper that documented the change in Arctic cloud biases in the ECHAM6 atmospheric model when ice supersaturation was allowed (Kretzschmar et al. 2018, published in ACP). The authors found a positive cloud cover bias when compared to CALIPSO due to an overestimation of low-level liquid-containing clouds, and attributed the bias to cloud microphysics. They were able to improve the phase partition between cloud liquid and ice by improving the Wegener-Bergeron-Findeisen process, but the cloud cover bias was only reduced when they allowed for slight supersaturation with respect to ice. Without having the specific information on which of the models in our study have ice supersaturation, the findings in Kretzschmar et al. mirror what we see in our analysis for the models that produce larger low cloud cover. These models have a much larger ice fraction and while one might expect that this leads to more precipitation/removal of ice and hence less cloud cover, other microphysical processes were found to overcompensate for this.

While we did not have complete information on which models allowed supersaturation w.r.t. ice, we do think it is a good suggestion to try grouping the models based on differences in the cloud microphysical parameterizations. Below (Fig. 3) are joint distributions in DJF of CA (first row), CLW (second row), CLI (third row), and ice condensate fraction (ICF, fourth row) for two new groupings of models: 1) calculate both cloud ice and cloud water as prognostic cloud variables, and 2) calculate a single mixing ratio of total water and use a temperature dependent partition to determine phase.

The first thing to notice about the above plots is a visualization of the process described in the previous paragraph whereby the models possessing higher ice fraction/ice mass actually have more cloud cover rather than less. Second, the models that calculate a mixing ratio of total water have less ice and more water than those that calculate both ice and liquid prognostically. For these models, the bounds of the

temperature-dependent partitions that determine ice vs. liquid vary. In between these boundary conditions are mixed-phase clouds, and individual model parameterizations determine the growth of ice via the Wegener-Bergeron-Findeisen process or heterogeneous freezing. Since the mixed-phase cloud regime is very common in the Arctic, and that relative concentrations of liquid and ice in the mixed-phase regime vary strongly for different model microphysics parameterizations, it is no surprise that the difference plots for CLW, CLI, and ICF (right column) between these two groupings of models are very large. The difference in cloud fraction between these groupings is smaller than that between the two model groups in our paper, indicating that differences in cloud microphysical schemes is part of the answer as to why the models simulate different clouds, but not the whole story.

6. There is minimal use of observations and comparison with observations in this work. It is hard to tell what is right, would like to see more comparisons against observations, and discussion and conclusions which focus on comparisons with observations. Which group is more like observations?

We too are interested in knowing which of these models or groups are "correct". However, this is a difficult question to answer and thoroughly addressing this question is beyond the scope of this paper. Our focus in this study is answering the question 'Why are the models low cloud amount annual cycle so different?'. A detailed observational comparison study is underway and will be part of a second paper using the same methodology (joint distribution analysis stratifying cloud amount by atmospheric state and cloud influencing factors) applied to observations and CALIPSO-CloudSAT satellite simulator output from available models. We have added a few sentences to the discussion about the observational comparison (Line 561-567).

Moreover, Referee #2 also indicated an interest in how the results might change if we used a different reanalysis dataset. To compare against the model output, the corresponding results based on one observational dataset (C3M) and 2 reanalysis datasets (MERRA-2 and ERA-Interim) are presented in the bottom panels of Figure 3.

Specific Comments:

Page 5, L164: I'm not sure I would say that the low cloud differences are spatially uniform. Differences seem lower over open water than sea ice for example, and largest differences are over land.

To address the reviewer's concern, we have calculated the average difference in the low and high clouds in DJF and JJA seasons between groups G1 and G2, for all gridpoints, land gridpoints, and ocean gridpoints separately. The results are below:

DJF Low Cloud Differences G1-G2all gridpoints = 12.02% G1-G2land only = 11.20% G1-G2ocean only = 12.57% DJF High Cloud Differences G1-G2all gridpoints = 6.38% G1-G2land only = 7.24% G1-G2ocean only = 5.81% JJA Low Cloud Differences G1-G2all gridpoints = -7.30% G1-G2land only = -6.56% G1-G2ocean only = -7.78% JJA High Cloud Differences G1-G2all gridpoints = 3.69% G1-G2land only = 3.34% G1-G2ocean only = 3.92%

From the above, one can see that CA differences between Group 1 and Group 2 are very similar whether you use all gridpoints, or ocean and land separately. In order to further quantify the effect of surface type, we have calculated the 95% confidence intervals for the difference in G1-G2 between land gridpoints vs all gridpoints and ocean gridpoints vs all gridpoints.

Results are below: DJF Low Cloud Differences G1-G2land only - G1-G2all gridpoints = 11.20% - 12.02% = -0.82% with a 95% CI of [-0.99, -0.65] G1-G2ocean only - G1-G2all gridpoints = 12.57% - 12.02% = 0.54% with a 95% CI of [0.39, 0.69] DJF High Cloud Differences G1-G2land only - G1-G2all gridpoints = 7.24% - 6.38% = 0.86% with a 95% CI of [0.81, 0.91] G1-G2ocean only - G1-G2all gridpoints = 5.81% - 6.38% = -0.57% with a 95% CI of [-0.61, -0.53] JJA Low Cloud Differences G1-G2land only - G1-G2all gridpoints = -6.56% - -7.30% = 0.74% with a 95% CI of [0.6, 0.87] G1-G2ocean only - G1-G2all gridpoints = -7.78% - -7.30% = -0.49% with a 95% CI of [-0.58, -0.4] JJA High Cloud Differences G1-G2land only - G1-G2all gridpoints = 3.34% - 3.69% =

-0.35% with a 95% CI of [-0.4, -0.29] G1-G2ocean only - G1-G2all gridpoints = 3.92% - 3.69% = 0.23% with a 95% CI of [0.2, 0.26]

In all months for all cloud types, the group difference G1-G2 between all gridpoints and land gridpoints/all gridpoints and ocean gridpoints is never more than 1% within the 95% confidence interval, which is much less than the average difference between Group 1 and Group 2. For this reason, we think it is appropriate to perform our calculations using all gridpoints. We added a comment about these results to lines 257-259.

Page 6, L191: shouldn't you do this by season (winter-summer) or at least comment on differences between winter and summer PDFs. Maybe show a sub set?

We constructed winter (DJF) and summer (JJA) PDFs for different cloud influencing factors:

DJF: (Figure 4)

And JJA:

While the average values of these quantities/shape of the PDF differ between DJF, JJA, and annual-mean, the difference between groups 1 and 2 remains consistent across different seasons: i.e., Group 1 models are relatively drier, have lower stability, larger ice fraction, and a smaller amount of liquid condensate compared to Group 2. This point has been added in the new text.

Page 6, L215: why are there vertical stripes here? Is this one model? Does it represent anything physical? The stripes result from two models, bcc-csm1-1 and NorESM1-ME, and do not represent anything physical, but are due to differences is the vertical resolution of these models relative to the others.

Reviewer Responses for Referee #2

1) Let us remind ourselves that we are in the Arctic, the region that has been chronically problematic not only for models, but also for observations and reanalysis datasets.

I can't help but wonder if the conclusions would change if the authors use ERAInterim/ERA5/JMA etc. instead of MERRA-2. Hinging their conclusions drawn from the stratification analysis (esp LTS, w) only on MERRA 2 is a bit risky.

We agree with you that reanalysis in the Arctic has significant problems, especially in the lower tropospheric temperature profile. However, we would like to clarify that the results of our analysis do not hinge on a reanalysis. The stratification analysis is performed with CMIP5 model output LTS, vertical velocity, and low cloud amount. In the future observational analysis the reanalysis used must be a prime consideration. To provide an additional reanalysis perspective, we have now included ERA-Interim in the analysis of Arctic cloud amount (e.g. Figs. 1 and 2).

2) The parameters like LWP and IWP have the largest uncertainties, no matter if you analyse reanalysis or observational data. How does this play a role? Also, can all models explicitly resolve cloud ice and cloud liquid water separately? Or does the partitioning depend on the temperature profile?

Referee #1 suggested that a table giving more details of the model microphysics schemes would be helpful, and we agree. We have added a new Table to provide a short description of cloud microphysics scheme. Many models do calculate cloud liquid and ice separately, while others calculate a single mixing ratio of total water, and use a temperature dependent partition to obtain liquid and ice. Both types of schemes are presented in Group 1 and Group 2, indicating that a model's specific microphysics scheme is not solely responsible for the seasonal cycle biases. For example, we may hypothesize that models that obtain cloud ice and liquid individually rather than a total condensate more accurately represent Arctic mixed phase clouds, but if these models also had too coarse a vertical resolution to resolve the supercooled liquid water layer, we would not see an improvement in the simulation of cloud amount. This is not to say that the way in which models treat cloud water phase is not important, only that the complexity of GCMs is such that only one parameterization alone cannot explain the cloud fraction differences we see. We found Komurcu et al 2014 ("Intercomparisons of the cloud water phase among global climate models") to be an informative resource; they studied the response of simulated cloud phase in GCMs to changes in ice nucleation schemes and found that implementing the same ice nucleation scheme in all of the models did not reduce the spread in cloud phase. In response to Referee #1, we grouped the models by those that prognostically calculate both ice and liquid, and those that calculate a single mixing ratio of total water and use a temperature dependent partitioning to determine cloud ice and liquid and plotted joint distributions of CA, CLI, CLW, and ice fraction. Please see the discussion on the differences in model parameterizations above in the response to reviewer #1.

3) Over the Arctic Ocean, what kind of biases in the annual cycles of cloudiness models show if they are stratified according to sea-ice conditions, for example, permanently sea-ice covered regions versus completely ice-free regions?

We have plotted seasonal cycles of cloud fraction for three surface types (Fig. 5) to address this comment: (Land: top left; Ocean: top right; Sea ice: bottom left)

The similarities in seasonal cycle biases between the three surface types include 1) the largest model spread occurring in winter, and 2) the same models with too few winter clouds over the entire domain also have too few winter clouds over each surface type (and vice versa for those models with too many winter clouds). The largest difference between the three surface types is found during summer, where land shows a smaller cloud fraction than either ocean or sea ice. Additionally, even though the general pattern of each models' seasonal cycle is similar across surface types, the seasonal amplitudes (winter versus summer) are greatest over sea ice and ocean and damped over land.

4) The differences in the representation of dynamical meteorology among models are also importing while interpreting the results. For example, do models show similar heat and moisture transport into the Arctic, which has a strong influence on cloudiness?

This is a very interesting and important question. Previous work (e.g., Morrison et al.

2012) highlights the important role that moisture advection plays in the maintaining low-level mixed phase clouds in the Arctic. Moreover, Boisvert et al. (2016) show the important effect that moisture transport by storms can have on Arctic sea ice and clouds. However, moisture advection/transport is a metric and process that we think is inadequately represented and potentially misrepresented by monthly averaged data. Therefore, we have decided to not include analysis of the influence of dynamics here. We recommend and will incorporate this comment into our future work, as we agree with you that atmospheric dynamics and moisture transport is a key consideration here. Addressing the role of dynamics requires the use of daily model output.

References Boisvert, L.N.; Petty, A.A.; Stroeve, J.C. The Impact of the Extreme Winter 2015/16 Arctic Cyclone on the Barents–Kara Seas. Mon. Weather Rev. 2016, 144, 4279–4287. Komurcu, M., Storelvmo, T., Tan, I., Lohmann, U., Yun, Y., Penner, J. E., Wang, Y., Liu, X., and Takemura, T. (2014), Intercomparison of the cloud water phase among global climate models, J. Geophys. Res. Atmos., 119, 3372– 3400, doi:10.1002/2013JD021119.

Kretzschmar, J., Salzmann, M., Mülmenstädt, J., and Quaas, J.: Arctic cloud cover bias in ECHAM6 and its sensitivity to cloud microphysics and surface fluxes, Atmos. Chem. Phys. Discuss., https://doi.org/10.5194/acp-2018-1135, in review, 2018.

Morrison, H.; de Boer, G.; Feingold, G.; Harrington, J.; Shupe, M.D.; Sulia, K. Resilience of persistent Arctic mixed-phase clouds. Nat. Geosci. 2012, 5, 11–17.

Please also note the supplement to this comment: https://www.atmos-chem-phys-discuss.net/acp-2018-1159/acp-2018-1159-AC1-supplement.pdf
* * *
[Figure]

|  | Ensemble Mean | Group 1 | Group 2 | Group 3 |
|---|---|---|---|---|
| Avg. DJF $CA$ (%) | 21.9% | 28.3% | 18.6% | 25.9% |

**Fig. 1.** Average winter low cloud fraction within LTS and omega500 regimes for the five models closest to observations.

[Figure]

**Fig. 2.** Joint distributions of average low cloud fraction binned by LTS and omega500 using daily data from the IPSL-CM5A-LR in winter

**Fig. 3.** Joint distributions of average low cloud fraction binned by LTS and omega500 for Group A and Group B

And JJA:

**Fig. 4.** Probability distributions of cloud influencing factors in winter and summer

**Fig. 5.** Seasonal cycle of cloud fraction by surface type (Land: top left; Ocean: top right; Sea ice: bottom left)

---

## Author Response (AR1)

**Reviewer Responses for Referee #1**

**1. The authors need to show that the results are robust to changes in model groups. Perhaps 1/3 of the models are very close to a 1:1 line they use to select models. What happens if you change the grouping of models? Does it change the results?**

As a first test of the model groups, we grouped the five models closest to the 1:1 line into a third group (hereafter Group 3) and constructed joint distributions of *CA* for this group (these models are bcc-csm1-1, CMCC-CM, CanESM2, MPI-ESM-MR, and MPI-ESM-LR; 2 Group 1 models and 3 Group 2 models). These models have smaller differences between average winter and summer *CA* compared to other models in their respective groups, thus we wouldn't expect the joint distributions for this group to resemble either Group 1 or Group 2 explicitly. Below is joint distribution for DJF for Group 3, to be compared to Fig. 8 in the paper. The table on the right shows average DJF *CA* for the ensemble, Group 1, Group 2, and Group 3.

The joint distribution for Group 3 contains features present in the joint distributions of both Group 1 and 2 as expected, given that Group 3 is made up of models from each group. For DJF, *CA* increases with increasing - $\omega_{500}$  for low-medium stability (similar to Group 2) but with larger average cloud amount (similar to Group 1). Also similar to Group 1 is the larger *CA* present at high stability and rising motion. The average values of *CA* from the table indicate that Group 1 *CA* > Group 3 *CA* > Group 2 *CA*, as expected for Group 3 given that it contains models from both Group 1 and 2. These examples are given to show that the 1:1 line separating Groups 1 and 2 is a good measure for group selection- if this were not the case then we might expect Group 3 joint distributions to resemble either Group 1 or Group 2. Since Group 3 joint distributions show features from both groups, this is an indication that even the models closest to the 1:1 still represent the low cloud responses of their respective groups. A small change in the grouping has a small effect on the results and does not affect our conclusions.

**2. The authors claim that since their results agree with earlier work, it is fine to use monthly data. That is not sufficient. They are averaging over regimes that may yield very different results, and they need to verify with a single model perhaps that monthly data for joint PDFs for example matches high frequency (daily or higher) data.**

To address this concern, joint distributions of low cloud amount binned by *LTS* and -*absob* were constructed using daily data from IPSL-CM5A-LR for winter (DJF) months and summer (JJA) months for the historical period 1979-2005. Additionally, eqn (1) from the paper was also

calculated using the daily data to confirm the validity of monthly data. The CMIP5 archive only had daily vertical cloud amount for one model available, (IPSL-CM5A-LR, a model from Group 1). The results from this model are presented below:

Equation 1:  $\overline{LCA} = \sum_{i,j} LCA(LTS_i, -\omega_{500,j}) * RFO(LTS_i, -\omega_{500,j})$ , describing the weighted sum of low cloud amount over *LTS* and  $-\omega_{500}$  from each i,j bin where  $LCA(LTS_i, -\omega_{500,j})$  is the low cloud amount as a function of *LTS* and  $-\omega_{500}$  and  $RFO(LTS_i, -\omega_{500,j})$  is the relative frequency of occurrence of each *LTS* and  $-\omega_{500}$  bin. Applying (1) to daily data from IPSL-CM5A-LR reproduces the domain-averaged LCA with the same accuracy as shown by monthly data in Table 2.

IPSL-CM5A-LR DJF domain-averaged LCA: 25.95% IPSL-CM5A-LR DJF LCA from Eq. (1): 25.91% IPSL-CM5A-LR JJA domain-averaged LCA: 16.6% IPSL-CM5A-LR JJA LCA from Eq. (1): 16.5%

Joint distributions for DJF (left) and JJA (right) low cloud amount binned by LTS and  $-\omega_{500}$  are shown below for IPSL-CM5A-LR constructed from daily data (top row) and monthly data (bottom row). Joint distributions for JJA look very similar between the daily and monthly versions: both show a strong gradient in LCA when LTS increases, and the largest LCA for high stability and rising motion. Additionally, the frequency of occurrence of LTS/-  $\omega_{500}$  regimes is similar when using either daily or monthly data. One difference between the JJA joint distributions is the presence of highly-stable regimes captured in the daily data (LTS > 25) that are not present in the monthly. However, these highly-stable regimes occur very infrequently (less than the 0.1% frequency of occurrence contour). Differences in DJF joint distributions are larger than for JJA. For low stability (LTS < 12), both daily and monthly distributions show LCA dependent on LTS with little dependence on - $\omega_{500}$ . For medium stability (12 < LTS < 26), both show similar amounts of low cloud (particularly in the most frequent regimes) but the daily data shows a slight gradient of larger LCA with increasing  $-\omega_{500}$  (this matches with the monthly joint distribution for Group 1 (Fig. 8a in the paper)). The largest differences between daily and monthly data occur for very high stability, as was the case for JJA. Daily joint distributions show the largest LCA for LTS > 34, particularly with rising motion. This is an infrequent regime, however, that the monthly distribution does not capture.

---

## Author Response (AR2)

**Reviewer Response Document – Reviewer Round 2**

Thank you very much. We have put a lot of effort in trying to make this manuscript as complete, informative, and useful as possible. Thank you for your time and effort in reviewing this manuscript.

**1. Grouping: I appreciate that the authors have tried another grouping strategy. I am not sure I follow the logic for a 'middle' group, but I think it proves their point. This needs to be mentioned in the text.**

The following text has been added to the manuscript to describe the robustness of our grouping strategy:

"As a test of the robustness of the grouping strategy, we created a third group containing the five models closest to the C3M observations (hereafter Group 3; bcc-csm1-1, CMCC-CM, CanESM2, MPI-ESM-MR, and MPI-ESM-LR). Composites of $CA$ for from Group 3 show features present in both Group 1 and Group 2, as expected since Group 3 contains models from each (not shown). This indicates that even the models closest to observations display features from their respective group. If the 1:1 line was a poor metric to use for group selection, we would expect Group 3 to resemble one of the groups or neither of the groups. Thus, the results are robust to a small change in the grouping strategy."

**2. Daily data v. Monthly data: I'm concerned that the DJF joint distributions are qualitatively different since with daily data there is an increase with omega 500 at constant LTS, and a monotonic increase in frequency at higher LTS for all omega 500, neither clear in the monthly data. I don' think that comment in the text is sufficient: There are some differences besides the dynamic range. Do they matter for the conclusions?**

**Please describe that the differences do not affect the conclusions, because they look like they might from this figure. I think this needs a little more work in the text, and maybe even inclusion of this figure to explain it.**

The following text has been added to the manuscript and have decided not to add the figure to the manuscript:

"The largest differences between the daily and monthly results occur in winter for high stability regimes ($LTS > 34$) in which daily data shows about 10% larger $CA$ than monthly; however, these regimes occur with a frequency less than 0.1%. We also note that the covariances between clouds and cloud influencing factors evaluated at daily and monthly timescales represent different manifestations of processes; thus, different processes may be important for explaining cloud behavior and model differences at the daily and monthly timescales. As such, care must be taken in the interpretation of the results at monthly timescales. We do not expect that the use of monthly averaged data to affect the main conclusions, however an analysis performed at the daily timescale provides more detailed information due to the larger dynamic range with the potential to identify additional processes that cause model differences under the wider range of atmospheric conditions."